# SIRIUS: Contextual Sparsity with Correction for Efficient LLMs

**Yang Zhou**[1], **Zhuoming Chen**[1], **Zhaozhuo Xu**[2], **Xi Victoria Lin**[3], **Beidi Chen**[1,3]
[1]Carnegie Mellon Univeristy
[2]Stevens Institute of Technology
[3]FAIR at Meta
{yangzho6, zhuominc, beidic}@andrew.cmu.edu
zxu79@stevens.edu victorialin@meta.com

## Abstract

With the blossom of large language models (LLM), inference efficiency becomes increasingly important. Various approximate methods are proposed to reduce the cost at inference time. Contextual Sparsity (CS) is appealing for its training-free nature and its ability to reach a higher compression ratio seemingly without significant performance degradation. However, after a comprehensive evaluation of contextual sparsity methods on various complex generation tasks, we find that although CS succeeds in prompt-understanding tasks, it significantly degrades the model performance for reasoning, deduction, and knowledge-based tasks. Despite the gap in end-to-end accuracy, we observed that sparse models and original models often share the general problem-solving logic and require only a few token corrections to recover the original model performance. This paper introduces SIRIUS[1], an efficient correction mechanism, which significantly boosts CS models on reasoning tasks while maintaining its efficiency gain. SIRIUS is evaluated on 6 models with 8 difficult generation tasks in reasoning, deduction, and coding and shows consistent effectiveness and efficiency. Also, we carefully develop a system implementation for SIRIUS and show that SIRIUS delivers theoretical latency reduction with roughly 20% reduction in latency for 8B model on-chip and 35% reduction in latency for 70B model offloading. We open-source our implementation of Sirius at `https://github.com/Infini-AI-Lab/Sirius.git`.

## 1 Introduction

Large Language Models (LLM), such as OpenAI et al. (2024) (GPT-4), Team et al. (2024) (Gemini), and Touvron et al. (2023) (Llama) have demonstrated their proficiency in a wide range of natural language processing applications such as content creation, summarization, and impressive and complex reasoning tasks. However, their deployment is very challenging, especially in latency-sensitive settings (Kaplan et al., 2020). Exploiting the model sparsity is a natural way to reduce the model parameter size and computational cost with a long history (LeCun et al., 1989; Tibshirani, 1996). More recently, many studies have shown that *contextual sparsity* (Liu et al., 2023b; Li et al., 2022; Dong et al., 2024; Lee et al., 2024), which highly correlates to the prompt or the context, can greatly speed up LLM inference without quality degradation.

However, in this paper, we first demonstrate a critical and fundamental problem with *contextual sparsity* (CS): while generally robust in classification tasks and generation tasks that mainly rely on prompt understanding (e.g., summarization, chat question-answering), we found that CS models struggle at high-level reasoning and understanding tasks.

---

[1]We draw inspiration from the astronomical concept, in which SIRIUS refers to a two-body star system, where one is the brightest star ever detected, while the other is a dim star.

38th Conference on Neural Information Processing Systems (NeurIPS 2024).

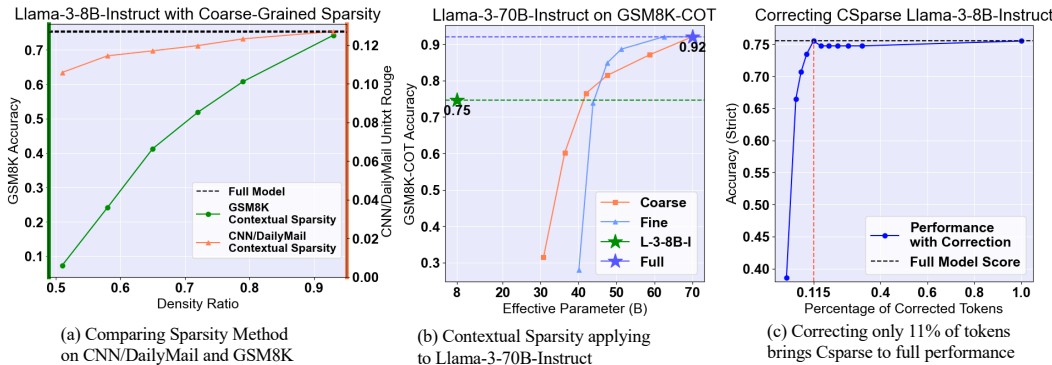

Figure 1: Contextual sparse models struggle at challenging text generation tests that require high-level reasoning and understanding, e.g. GSM8K. On these tasks, contextually sparse models lead to significant quality degradation. In (a), we contrast CS Llama-3-8B-Instruct on GSM8K (green) and CNN DailyMail (coral). (b) Contextual Sparsity Llama-3-70B-Instruct crashes at 50% global sparsity, making the smaller dense model Llama-3-8B-Instruct (green star) a significantly more efficient choice than the sparse 70B model. (c) Sparse model crashing at reasoning tasks has patterns, and ideally only correcting 11% unlikely tokens recovers the sparse model performance fully.

For example, in Figure 1 (a), we contrast between the Text Summarization task (CNN/DailyMail) and Arithmetic Reasoning (GSM8K) with contextual sparsity methods on Llama-3-8B-Instruct. Varying sparsity levels, Llama-3-8B-Instruct with contextual sparsity performs consistently worse on GSM8K than CNN/DailyMail. With roughly 50% sparsity globally, the sparse model degradation is still reasonable on text summarization (right axis in color coral) compared to almost collapsed on arithmetic reasoning (left axis in color green). **However, for reasoning tasks, can we simply live with a higher density ratio to preserve more performance?** Unfortunately, the answer is NO. Besides the significant efficiency loss, shown in Figure 1 (b), the Llama-3-70B-Instruct model with contextual sparsity also crashes at around 50% sparsity globally. The 50% sparse model still has 4× the parameter size compared to the smaller dense model (Llama-3-8B-Instruct), while still performing worse on GSM8K-CoT, rendering contextual sparsity utterly not useful for complex reasoning tasks.

We conduct an in-depth study on the CS model failure cases and notice that the overall reasoning pathway of these sparse models is usually sound and adheres to the full model. The fatal mistakes are always caused by some middle tokens and propagate towards the end, examples can be seen in Figure 4. Following this observation, we conduct a simple experiment with 65% Contextual Sparse Llama-3-8B-Instruct on GSM8K as presented in Figure 1 (c). We run both the sparse and the full models together for the same prompt and compare two generation output token-by-token.

Surprisingly, the trend increases steeply with the percentage of corrected tokens. Correction of only 6% tokens in the sparse model's generation recovers most of GSM8K accuracy (<5% to full), and 11% to recover the full performance. The results show potential for an efficient and powerful correction mechanism to maintain the sparse efficiency while boosting its performance. Contextual sparsity uses a dynamic sparsity pattern and naturally requires the full model to be in GPU memory during runtime, allowing the full model to be used efficiently for infrequent correction. Even though only very few need to be corrected, locating these mistaken tokens efficiently turns out to be challenging.

Ideally, we want a correction system to have the following properties: 1) **Effective**, the sparse model quality degradation can be improved to the full model vicinity; 2) **Cheap**, the full model only gives minimal intervention; 3) **Adaptive**, the system is efficient across various reasoning datasets.

In this paper, we carefully analyze and formulate correction efficiency in Section 2. We extensively categorize the strengths and weaknesses of CS in Section 3. In Section 4, We systematically design Sirius, a correction method covering all three desired properties. (**When?**) In Section 4.1, we show that the sparse model can be both confident or uncertain when making mistakes, rendering the signal from sparse unreliable for determining when to correct. Sirius is a period-based method with the period as a hyperparameter. (**How?**) In Sections 4.1 and B.2, we introduce novel KV Cache direct rewriting, minimal rollbacks, and hardware-efficient tree building to help increase the effective period of full model correction, thus, ensuring the correction efficiency.

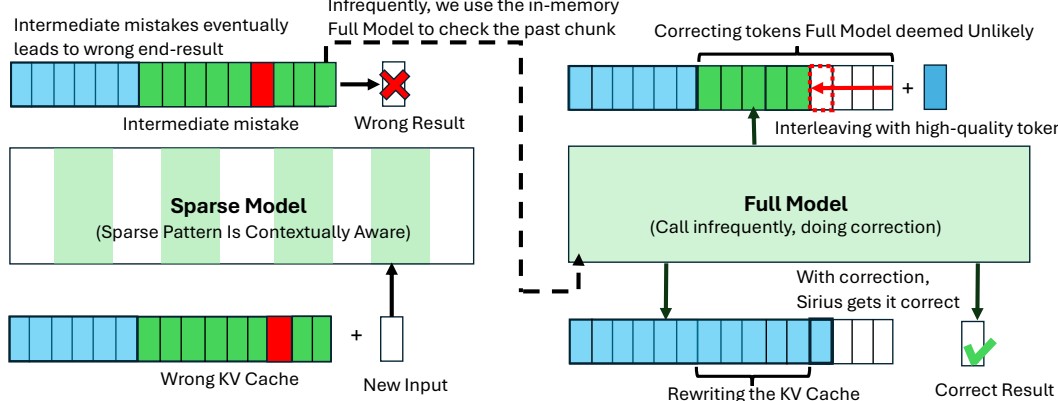

Figure 2: Overview of Sirius. Contextual Sparsity requires full model weights to be placed on the GPU memory. While the sparse model doesn't perform well on complex reasoning tasks, Sirius uses the Full Model to correct the Sparse model. The full model is called fairly infrequently. During the correction, the Full Model will rewrite the KV Cache, interleave with high-quality tokens to the sparse outputs, and then roll back only when the token is deemed extremely unlikely by the Full Model.

In Section 5, we empirically evaluated SIRIUS on 6 different models with 8 different reasoning tasks and showed that SIRIUS is generally effective and efficient. On GSM8K and Llama-3-8B-Instruct specifically, we boost the fine-grained sparsity from 58% to 72% with 4% increase in effective parameter size and coarse-grained sparsity from 38% to 70% with the cost of 5% effective parameter size. We also show that SIRIUS delivers the promised efficiency on mainstream GPUs in both on-chip and offloading settings.

## 2    Related Works and Problem Formulation

In this section, we first present the classification of the prior Contextual Sparsity methods and narrate important efficiency metrics in. **Also, we present careful analysis and quantitative comparisons on why Speculative Decoding is inefficient in recovering contextual sparsity.** Due to space constraints, we refer to Appendix A.2. For extended related works on model compression, contextual sparsity, and speculative decoding, we present in Appendix A.1.

**Contextual Sparsity Classification** - Contextual sparsity (CS) methods are usually training-free, easy to use, and seemingly effective, making them highly attractive to ML practitioners looking to reduce LLM inference costs. CS exists naturally in MLP layers of the LLM, which occupies roughly 70% of the LLM total weights (Dong et al., 2024; Lee et al., 2024)). The contextual sparsity selection is as follows: given the context, only a limited number of the most relevant neurons are selected based on the input activation. The rest contributed to the output far less is discarded. We refer to two main directions of contextual sparsity methods as **Coarse-grained Sparsity** (CSparse) Methods (Dong et al. (2024)) - that within the same input prompt, the sparsity pattern is fixed for all tokens generated. **Fine-grained Sparsity** (FSparse) Methods (Lee et al. (2024)) - that exploits the per-token sparsity to save resources.

**Average Parameters Used Per Token** - A key metric is used to evaluate the efficiency of our proposed method, the Average Parameter Used per token decoded (later referred to as APU). LLM inference is memory I/O bound (Leviathan et al., 2023; Kim et al., 2023). The latency of generating every single token is dominated by the memory loading time from the GPU HBM to SRAM. On the other hand, SIR-IUS relies on full model parallel verifying a chunk of tokens. Although from the FLOPs standpoint, the amount of compute performed per evaluation step is the number of input token times of a single token input process, the latency of parallel verification is still roughly the same as taking a single token (Verified further in 10, length 64 is only 1.1 ms longer than length 1), because the inference is memory bound.

SIRIUS operates in the memory-bound regime (single inference sequence length smaller than or equal to 64). Thus, the average parameter count of a model gives us a rough judgment of the latency of inference. Formally, for a full LLM to have $C_{full}$ number of parameters, and its sparse counterpart of a certain predetermined sparsity $C_{sparse}$. The average advancement length (later we refer to as AAL) in the number of tokens between two consecutive LLM corrections can be represented as $n_{AAL}$.

The average parameters used per token (APU) are the following

$$\text{APU} = \frac{n_{sparse}C_{sparse} + C_{full}}{n_{AAL}} \tag{1}$$

We want the metric to be as small as possible, and obviously, we want $n_{AAL}$ to be as large as possible.

Another thing to note is that we always compare the system's APU against the full model's APU, which is $C_{full}$. If we divided the above equation by $C_{full}$, we can have an equivalent parameter density of the system defined based on $I_{globalsparsity}$, which is $C_{sparse}/C_{full}$.

$$\text{Effective Density} = \frac{n_{sparse}I_{globalsparsity} + 1}{n_{AAL}} \tag{2}$$

Later, if we use period $n_{period}$, the equation can be rewritten as

$$\text{Effective Density} = \frac{(n_{period} - 1)I_{globalsparsity} + 1}{n_{AAL}} \tag{3}$$

Later when presenting SIRIUS, we mainly specify $n_{period}$ with $n_{AAL}$ to evaluate its efficiency. Notice that $I_{globalsparsity}$ is determined by the sparsity method, SIRIUS cannot change it anymore.

## 3 Observations

In this section, we present a detailed study of the strengths and weaknesses of Contextual Sparsity (CS). 3.1 presents the strengths of CS. 3.2 presents the weaknesses of CS. Additionally, we show that given the similar parameter size, the more well-trained the model is, the more CS degradation will be for the model. Due to limited space, we present the details in Appendix B.1. 3.3 shows our findings when looking into the failure cases of the CS model in complex reasoning generation tasks.

In the following series of experiments, we build our implementation[2] of fine-grained sparsity based on Lee et al. (2024) and coarse-grained sparsity based on Dong et al. (2024). The default sparsity for both methods is 50% for the MLP component of the model (whole MLP for coarse-grained sparsity and Up and Down linear layers only for fine-grained sparsity). We mainly use this default setting in most experiment tables in the paper without explicitly mentioning it. Otherwise, we will explicitly specify the different sparsity levels we used.

### 3.1 Contextual Sparsity: Where Does It Succeed?

For tasks on prompt understanding, CS generally performs well and gives consistent and strong output. We evaluate CS models on machine summarization (CNNDailyMail, See et al. (2017)), and Conversational Question Answering (CoQA, Reddy et al. (2019)). The results show that the correctly selected contextual sparsity in the MLP layers and the full attention layers can fully extract and understand the local prompt information. More details are presented in Figure 3, where we show that by varying the sparsity level, the language model's performance on CNN/DailyMail is robust even when the activation sparsity drops to below 20%, which translates to around 44% global density.

For tasks accessing factuality and hallucination, we select the generation portion of the TruthfulQA dataset (Lin et al., 2022). Results are shown in Table 1, where we evaluate the techniques on 5 different LLMs. Interestingly, we find that the Fine-grained sparsity is often better than the dense model baseline across different models. This finding is consistent with previous works Laser (Sharma et al., 2023) and Dola (Chuang et al., 2024). They both observed that compressing the original LLM in a carefully designed way would lead to improvement in factuality and better de-hallucination. Laser comes from the low-rank approximation of the MLP layers, while Dola proposes a factuality-aware layer-skipping algorithm. Based on their findings, hallucination occurs when parts of the weights aren't as well-versed in the given input as the other parts. They expose the "averaging" effect that blurs the factuality of the output. Removing these neurons gives rise to better facutality and less hallucination. Our studies look at the same problem from a neuron sparsity standpoint.

---

[2]Since Lee et al. (2024) doesn't open-source its implementation and it relies on the threshold for determining the sparsity pattern, replicating the method isn't straightforward. Using a threshold also increases the difficulty of determining the actual density of the sparse model. Our implementation uses topk on the Gate Layer activations. The rest is implemented as described in the original method.

Table 1: We show the difference between cases when Contextual Sparsity (CS) succeeds or fails. CS is generally good at prompt understanding tasks and tasks that measure the trustworthiness of the language models while not good at tasks that require reasoning and world knowledge understanding.

| Where CS Succeeds
Experiment Settings | CNN/DailyMail
Unitxt Rouge | CoQA
EM/F1 | TruthfulQA
Rouge-1/2 ACC |
|---|---|---|---|
| Llama-3-8B-Instruct | 0.1237 | 0.6153/0.7825 | 0.4945/0.3647 |
| Llama-3-8B-Instruct-CSparse | 0.1144 | 0.6633/0.7977 | 0.4725/0.3403 |
| Llama-3-8B-Instruct-FSparse | 0.1166 | 0.6625/0.7984 | 0.5043/0.3305 |
| Llama-2-7B-Chat | 0.1489 | 0.5982/0.7580 | 0.4480/0.3831 |
| Llama-2-7B-Chat-CSparse | 0.1448 | 0.6117/0.7639 | 0.4529/0.3843 |
| Llama-2-7B-Chat-FSparse | 0.1521 | 0.5898/0.7540 | 0.4565/0.3660 |
| Where CS Fails
Experiment Settings | GSM8K
ACC (strict/flexible) | HumanEval
Pass@1 (GD) | MMLU[*]
Accuracy |
| Llama-3-8B-Instruct | 0.7551/0.7544 | 0.560 | 0.6231 |
| Llama-3-8B-Instruct-CSparse | 0.3859/0.3874 | 0.207 | 0.5558 |
| Llama-3-8B-Instruct-FSparse | 0.5868/0.5891 | 0.457 | 0.5304 |
| Llama-2-7B-Chat | 0.2396/0.2462 | 0.140 | 0.492 |
| Llama-2-7B-Chat-CSparse | 0.1334/0.1380 | 0.067 | 0.4637 |
| Llama-2-7B-Chat-FSparse | 0.1979/0.2017 | 0.134 | 0.4768 |

[*] **MMLU** is a classification task, not generation tasks. We use **MMLU-FLAN-COT**

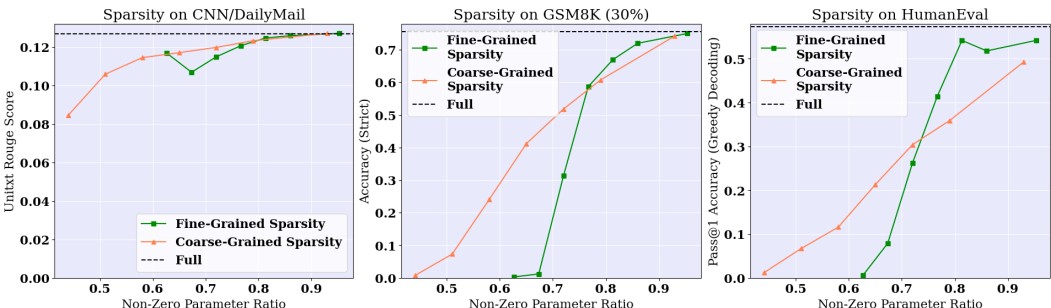

Figure 3: We contrast between Contextual Sparsity on prompt understanding task and complex generation tasks that require reasoning. (a) Both CSparse and FSparse are robust on CNN/DailyMail for various sparsity; (b) and (c) Show that both CSparse and FSparse crash on GSM8K and HumanEval at the global sparsity that they are still robust in prompt understanding tasks.

## 3.2 Contextual Sparsity: Where Does It Fail?

On the other hand, contextual sparsity severely struggles when the generation tasks rely solely on the model's own reasoning and world knowledge understanding ability. Here we show the Llama-3-8B-Instruct and the Llama-2-7B-Chat models in Table 1, refer to Table 12 for evaluations on more models. Notice that since fine-grained sparsity method needs the activation from Gate MLP for selecting sparsity, while coarse-grained sparsity has a predetermined pattern after prefilling and can sparsify the Gate MLP. Even though both are at 50% activation sparsity, the coarse-grained sparsity method effectively achieves higher parameter savings than fine-grained sparsity in practice. Here we evaluate the sparse techniques using 5-shot CoT on the GSM8K dataset (Cobbe et al., 2021). We found that across all the models we evaluated, both sparsity methods lead to significant accuracy degradation. We include HumanEval (Chen et al., 2021), a coding task that requires complex reasoning and planning ability. We found that both sparsity methods exhibit similar performance degradation when it comes to coding. Shown in Figure 3, two tasks see sparsity significantly drop performance after 50% activation sparsity.

For knowledge recall and world knowledge understanding, we specifically test on MMLU-Flan-CoT (Chung et al., 2022) the CoT text generation version of the MMLU dataset (Hendrycks et al., 2021). Table 1 shows the results. Stronger models like Llama-3-8B-Instruct suffer from significant

degradation too. Furthermore, we found that given the similar parameter size, the more well-trained the models are, the higher its degradation from the contextual sparsity, more details in Appendix B.1.

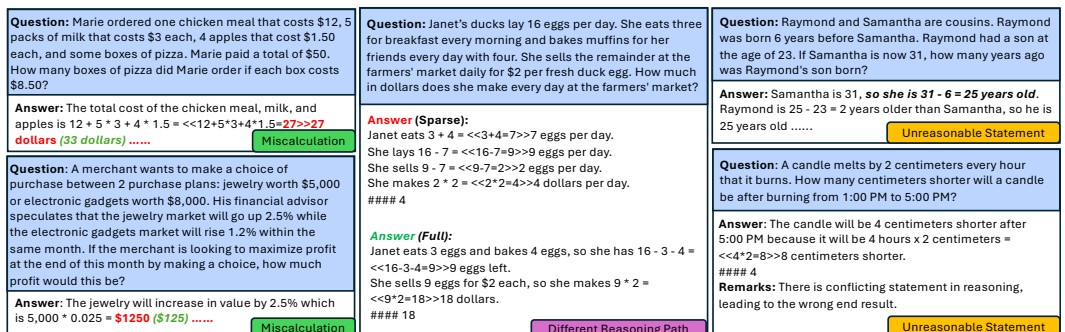

Figure 4: Examples of contextual sparse model making the identified three different types of mistakes. Most mistakes occur because the model makes calculation mistakes or has a wrong reasoning step compared to the full model. We also observe that there are rare cases where the model makes insensible statements in the middle that make the end result wrong.

### 3.3 A Closer Look on GSM8K Quality Degradation

To study the inability of the sparse model in deduction, we conduct a case study on the sequence-level coarse-grained sparsity methods Dong et al. (2024) with the Llama-3-8B-Instruct model. We visually inspect extensive cases where the sparse model and dense differ in answers. Generally, the sparse model always produces highly similar answers to the dense model: the similar approach or logic flow when approaching the same problem and even the same number of sentences before the first mistake occurs or in success cases. However, the key differences are usually caused by the following three categories of small token-level mistakes: (1) frequent miscalculation in the intermediate steps, (2) wrong reasoning in intermediate steps, and (3) insensible and random statements. For each of the above-summarized cases, we find failure question-answer pairs provided in Figure 4. These mistakes happen in the middle of arguments and propagate to the wrong end result.

Similar observations can also be found for fine-grained sparse methods with different model types. *Interestingly, we find that even with these mistakes, the sparse model can still fully generate coherent tokens and make further reasoning assuming their prior steps are correct.*

We hypothesize that the gap between the full model and these sparse counterparts is at these key tokens. The following simple experiment is conducted to further verify our hypothesis. We run the coarse-grained sparse model and the full model with the same input prompt and for every token the sparse model generates, the full model is used to check the likelihood of these decoded tokens, mainly removing tokens with low likelihood. By varying the likelihood threshold, we can control the frequency of the correction. The experiments are conducted for both Llama-3-8B-Instruct and Llama-2-7B-Chat Touvron et al. (2023) models with coarse-grained sparsity. The results are shown in Figure 1(c). In both cases, we found that a very small amount of correction would drastically improve the sparse model performance, showing a steep gradient when the percentage of corrected tokens is small. With merely 10% of tokens needing to be corrected, the sparse model can completely match the full model's performance. The experiment verifies our hypothesis that by correcting the small portion of key tokens, the sparse model can meet the large model's performance.

## 4 Methods

Though we find a minor portion of tokens needed to be corrected for the contextual sparsity model to fully recover performance, the challenge remains: how to locate these mistaken tokens with the minimal number of parallel verification rounds of the full model? In this section, we show that the sparse model provides signals that cannot be trusted 4.1. Then, we describe in detail the various correction techniques in 4.1. Because of the space limit, we put how to boost the sparse generation with hardware-efficient tree building B.2.

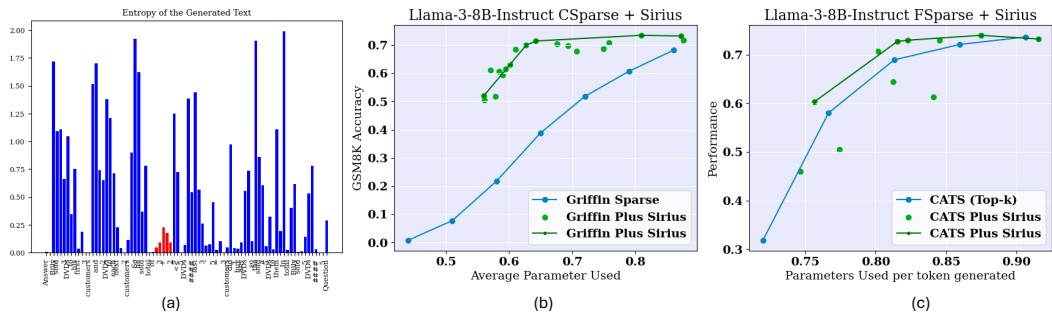

Figure 5: In (a), we present an example that illustrates why the signals from the sparse model are unreliable. It is a figure plotting entropy versus generated tokens. At the tokens where the sparse made the mistake (red), the entropy isn't in large spikes which signifies chaos and low confidence, rather it is even quite low, compared to nearby entropy spikes. In (b) and (c), we view Sirius as a compression method by itself. We compare Sirius with contextual sparse methods and show that given the same parameter used, Sirius performs better than Contextual Sparse Methods on GSM8K.

## 4.1 Sparse Model's Self-Awareness Cannot Be Trusted

Intuitively, rather than fixing the $n_{sparse}$ number, letting the system decide when to call the LLM for evaluation would then give more flexible $n_{sparse}$. Nevertheless, we argue that the sparse model's output probability distribution cannot be used as a metric for accuracy decisions. We empirically experiment with various methods to utilize the information contained in the sparse model's output distribution. However, varying the threshold leads to $n_{sparse}$ being too short when the threshold is strict or failing to correct when the threshold is lenient. We then discovered that the sparse model has very limited self-awareness of its own mistakes. To make the observation concrete, we present a small example in Figure 6 a piece of text where the sparse model makes a mistake while the full model succeeds. The red bars signify the error location. The token entropy is neither high nor at zero, making it impossible to effectively use a threshold to control the number $n_{sparse}$.

## 4.2 How to Correct the Sparse Output Tokens

The full overview of SIRIUS is presented in Figure 2 and Algorithm 1. The full model is called once every kernel size. The KV cache is shared between the sparse and the full model. The KV cache is mostly populated by the sparse model, which is called for every token. During correction, the full model takes in the last kernel size of tokens and generates its KVs for the past kernel size tokens in parallel, these KVs are directly written to their corresponding positions in the shared KV Cache. Empirically, we found that full model's KV helps the sparse model's output. When LLM is called to evaluate the sparse model's output, it uses its own predicted likelihood to determine whether to accept or reject the sparse model's past output. The decision is based on comparing the likelihood against a preset threshold. Detailed ablation for threshold is in B.3. Besides the above-mentioned techniques, we also found that the "second/third" choices of the sparse models' rejected token position offer > 80% coverage of the LLM accepted tokens. The observation motivates us to build a hardware-friendly tree on the sparse model generating side that doesn't sacrifice the performance while significantly boosting the $n_{AAL}$ or efficiency. Due to the space limit, a great amount of details is in Appendix B.2.

## 5 Experiments

In this section, we empirically evaluate SIRIUS to correct CS models on various generation tasks in complex reasoning. We show that SIRIUS is consistent in various tasks, effective in helping CS models recover their performance, and efficient in correction with low additional overhead.

- In 5.1, we evaluated SIRIUS on six models with 8 different datasets. SIRIUS is consistently effective and efficient. Specifically, on GSM8K, SIRIUS corrects FSparse Llama-3-8B-Instruct from 58% accuracy to 72% with only 4% increase in parameter density and corrects CSparse model from 38% to 70% with 5% density.
- In 5.2, we presents more details on our system implementation for SIRIUS. We show that SIRIUS delivers its theoretical efficient promise, achieving roughly 20% reduction in latency compared to full on-chip on various hardware. SIRIUS further achieves 35% speedup to full in offloading settings.

**Algorithm 1** Sirius

**Require:** Prompt $[x_1,...,x_t]$, full model $M_F$, and sparse model $M_S$ sharing weights and sharing KV Cache $C$, $cache\_pos$ is the location where new k and v are written to $C$, kernel size $n$

**Require:** forward function FORWARD, threshold $r$, which is a value used by $M_F$ to judge whether the token occurs likely enough

**Require:** StoppingCriteriaMet() downstream task-specific, returns a boolean

1: **while** not StoppingCriteriaMet() **do**
2:  $\quad i \leftarrow 0$
3:  $\quad kernel \leftarrow$ empty
4:  $\quad cache\_pos \leftarrow 0$
5:  $\quad$ **while** $i < n$ **do**
6:  $\quad\quad$ set $\hat{p_{t+i}} \leftarrow$ FORWARD$(M_C, C, [x_1,...,x_t], [x_{t+1}, ..., x_{t+i-1}], cache\_pos)$
7:  $\qquad\qquad\qquad\qquad\qquad\qquad\qquad\qquad\qquad\qquad$ ▷ Running sparse model
8:  $\quad\quad cache\_pos \leftarrow cache\_pos + 1$
9:  $\quad\quad$ sample $\hat{x_{t+i}} \sim \hat{p_{t+i}}$
10: $\quad\quad kernel \leftarrow$ cat$(kernel, \hat{x_{t+i}})$
11: $\quad\quad i \leftarrow i+1$ $\qquad\qquad\qquad\qquad\qquad$ ▷ Before exiting, $kernel\ [x_t,...,x_{t+n}]$
12: $\quad$ **end while**
13: $\quad cache\_pos \leftarrow cache\_pos$ subtracts n $\qquad$ ▷ Enables Full to directly rewrites KV Cache
14: $\quad$ set $[q_t,...,q_{t+i}] \leftarrow$ FORWARD$(M_C, C_S, cache\_pos, kernel)$
15: $\quad$ **for** $j$ from 0, n **do**
16: $\quad\quad$ **if** $q_{t+j} < r$ **then** $\qquad\qquad\qquad\qquad\qquad\qquad\qquad$ ▷ Full rejects
17: $\quad\quad\quad$ break $\qquad\qquad\qquad\qquad$ ▷ $j$ stores the first token position being rejected
18: $\quad\quad$ **end if**
19: $\quad$ **end for**
20: $\quad cache\_pos \leftarrow j+1$ $\qquad\qquad\qquad\qquad\qquad\qquad\qquad$ ▷ Rollback
21: $\quad kernel \leftarrow$ empty
22: $\quad$ sample $x_{t+j+1} \sim p_{t+j}$ $\qquad\qquad\qquad\qquad\qquad$ ▷ Interleaving Key Token
23: **end while**

- We also present ablation with rich details on how each component of SIRIUS contributes to its performance and how threshold is used to trade off efficiency and performance. Due to space limit, we place it in Appendix B.3.

### 5.1 Sirius Significantly Recovers CS Degradation with Low Cost

**Models and Datasets** - To comprehensively evaluate SIRIUS performance, we deploy six mainstream LLMs with sizes ranging from 7B to 13B: Llama-2-7B, 13B, and Llama-3-8B with their instruction finetuned counterparts, all from Llama family. Following Wei et al. (2022) in LLM reasoning, we also tested CS models on two popular types of reasoning generation tasks: arithmetic and commonsense reasoning. On the Arithmetic side, besides GSM8K, we also evaluate CS models on AQuA-RAT. On the Common Sense side, we use CSQA Saha et al. (2018), StrategyQAGeva et al. (2021), Date, and Sports, last two from Big Bench Suite bench authors (2023). Most of these tasks are originally classification tasks. Following the instruction in Wei et al. (2022), we manually compose COT prompts to transform these into logic argument generation tasks. Besides, CS models do not perform well in coding, which requires forming logical arguments and planning. We select HumanEval Chen et al. (2021) and MBPP+ Liu et al. (2023a) to evaluate SIRIUS.

For arithmetic reasoning and coding, we use 50% neuron sparsity for both CSparse and FSparse. FSparse relies on the gate layer to be dense, leading to higher global density than CSparse. Since commonsense reasoning tasks are generally less logically challenging comparatively, we lowered the neuron sparsity level to 40%.

**Main Results** - Due to space limits, we only select the best treewidth of SIRIUS for GSM8K, CSQA, and HumanEval for the main results in Table 2. Extensive studies on the rest 5 datasets with different treewidth are presented in the Appendix C. From Table 2, we can see that SIRIUS is consistently effective and efficient across all different classes of tasks. Specifically for Llama-3-8B-Instruct, besides GSM8K, SIRIUS corrects FSparse and CSparse, on CSQA, from 61% and 64% accuracy to 70% with cost only 3% sparsity for FSparse and 7% for CSparse respectively. On HumanEval, SIRIUS corrects FSparse from 45% to 61% with 4% sparsity overhead even surpassing the full model's performance,

Table 2: We show SIRIUS effectiveness and efficiency in the following table. We select GSM8K for Arithmetic Reasoning, CSQA for Commonsense Reasoning, and HumanEval for code generation. Under the "SIRIUS Perf. " column, A(B) is shown. A denotes the accuracy after SIRIUS correction in the dataset evaluated, while (B) represents the optimal treewidth selected under the current model dataset settings. Under the column of "AAL", X/Y is shown, where X is the AAL, while Y is the period.

| GSM8K | | | | | | |
|---|---|---|---|---|---|---|
| **Model** | **Full Perf.** | **CSparse Perf.** | **CSparse Density** | **SIRIUS Perf.** | **AAL** | **Effective Density** |
| Llama-3-8B-Instruct | 0.7536 | 0.3844 | 0.65 | 0.7051 (8) | 15.22/16 | 0.706 |
| Llama-3-8B | 0.4966 | 0.2085 | 0.65 | 0.4177 (8) | 15.29/16 | 0.703 |
| Llama-2-7B-Chat | 0.2403 | 0.1334 | 0.69 | 0.2244 (8) | 15.00/16 | 0.757 |
| Llama-2-7B | 0.1357 | 0.0758 | 0.69 | 0.1183 (6) | 15.87/16 | 0.715 |
| Llama-2-13B-Chat | 0.3548 | 0.2714 | 0.68 | 0.3381 (4) | 15.34/16 | 0.730 |
| Llama-2-13B | 0.2282 | 0.1759 | 0.68 | 0.2418 (1) | 15.34/16 | 0.730 |

| **Model** | **Full Perf.** | **FSparse Perf.** | **FSparse Density** | **SIRIUS Perf.** | **AAL** | **Effective Density** |
|---|---|---|---|---|---|---|
| Llama-3-8B-Instruct | 0.7536 | 0.5868 | 0.76 | 0.7278 (4) | 15.37/16 | 0.807 |
| Llama-3-8B | 0.4966 | 0.3199 | 0.76 | 0.4579 (2) | 15.03/16 | 0.825 |
| Llama-2-7B-Chat | 0.2403 | 0.1971 | 0.79 | 0.2388 (6) | 15.69/16 | 0.819 |
| Llama-2-7B | 0.1357 | 0.1137 | 0.79 | 0.1410 (4) | 15.91/16 | 0.807 |
| Llama-2-13B-Chat | 0.3548 | 0.3222 | 0.78 | 0.3533 (1) | 15.08/16 | 0.842 |
| Llama-2-13B | 0.2282 | 0.2191 | 0.78 | 0.2372 (4) | 15.92/16 | 0.797 |

| CSQA | | | | | | |
|---|---|---|---|---|---|---|
| **Model** | **Full Perf.** | **CSparse Perf.** | **CSparse Density** | **SIRIUS Perf.** | **AAL** | **Effective Density** |
| Llama-3-8B-Instruct | 0.7073 | 0.6470 | 0.58 | 0.7076 (8) | 14.76/16 | 0.657 |
| Llama-3-8B | 0.6437 | 0.5585 | 0.58 | 0.6429 (8) | 15.43/16 | 0.628 |
| Llama-2-7B-Chat | 0.6248 | 0.5200 | 0.62 | 0.6175 (8) | 15.07/16 | 0.683 |
| Llama-2-7B | 0.4742 | 0.4414 | 0.62 | 0.4742 (8) | 15.80/16 | 0.652 |
| Llama-2-13B-Chat | 0.6879 | 0.5536 | 0.61 | 0.6691 (4) | 11.43/12 | 0.674 |
| Llama-2-13B | 0.6109 | 0.5601 | 0.61 | 0.6060 (4) | 15.72/16 | 0.645 |

| **Model** | **Full Perf.** | **FSparse Perf.** | **FSparse Density** | **SIRIUS Perf.** | **AAL** | **Effective Density** |
|---|---|---|---|---|---|---|
| Llama-3-8B-Instruct | 0.7073 | 0.6158 | 0.72 | 0.7043 (8) | 15.66/16 | 0.753 |
| Llama-3-8B | 0.6437 | 0.533 | 0.72 | 0.6388 (1) | 15.00/16 | 0.786 |
| Llama-2-7B-Chat | 0.6248 | 0.6167 | 0.75 | 0.6380 (4) | 15.09/16 | 0.811 |
| Llama-2-7B | 0.4742 | 0.4717 | 0.75 | 0.5012 (6) | 15.89/16 | 0.771 |
| Llama-2-13B-Chat | 0.6879 | 0.533 | 0.74 | 0.6691 (4) | 14.30/16 | 0.846 |
| Llama-2-13B | 0.6109 | 0.5700 | 0.74 | 0.5864 (4) | 15.72/16 | 0.770 |

| HumanEval | | | | | | |
|---|---|---|---|---|---|---|
| **Model** | **Full Perf.** | **CSparse Perf.** | **CSparse Density** | **SIRIUS Perf.** | **AAL** | **Effective Density** |
| Llama-3-8B-Instruct | 0.561 | 0.207 | 0.65 | 0.524 (8) | 14.67/16 | 0.733 |
| Llama-3-8B | 0.262 | 0.067 | 0.65 | 0.243 (8) | 15.10/16 | 0.691 |
| Llama-2-7B-Chat | 0.140 | 0.067 | 0.69 | 0.159 (8) | 10.88/12 | 0.789 |
| Llama-2-7B | 0.116 | 0.079 | 0.69 | 0.128 (8) | 14.84/16 | 0.765 |
| Llama-2-13B-Chat | 0.189 | 0.122 | 0.68 | 0.171 (8) | 11.12/12 | 0.762 |
| Llama-2-13B | 0.262 | 0.067 | 0.68 | 0.244 (8) | 15.10/16 | 0.741 |

| **Model** | **Full Perf.** | **FSparse Perf.** | **FSparse Density** | **SIRIUS Perf.** | **AAL** | **Effective Density** |
|---|---|---|---|---|---|---|
| Llama-3-8B-Instruct | 0.561 | 0.457 | 0.76 | 0.616 (6) | 15.42/16 | 0.804 |
| Llama-3-8B | 0.262 | 0.189 | 0.76 | 0.298 (6) | 15.54/16 | 0.797 |
| Llama-2-7B-Chat | 0.140 | 0.134 | 0.79 | 0.165 (6) | 15.27/16 | 0.841 |
| Llama-2-7B | 0.116 | 0.116 | 0.79 | 0.165 (6) | 15.86/16 | 0.810 |
| Llama-2-13B-Chat | 0.189 | 0.146 | 0.78 | 0.183 (6) | 15.34/16 | 0.827 |
| Llama-2-13B | 0.246 | 0.233 | 0.78 | 0.259 (4) | 15.85/16 | 0.801 |

and from 20% to 52% with 8% sparsity as cost. Besides, Llama-3-8B-Instruct, SIRIUS corrects all 6 models with additional sparsity overhead smaller than 10% across these three datasets, further showing its strong efficiency. Besides results in Table 2, in Appendix C, we show that SIRIUS consistently shows great effectiveness with high efficiency across the rest of the 5 datasets.

## 5.2  Wallclock Speedup

Table 3: Performance and Speedup Ratios on GSM8K-COT with Different Hardware Configurations.

| Settings | ACC | A40 | Ratio | L40 | Ratio | A100 | Ratio | H100 | Ratio |
|---|---|---|---|---|---|---|---|---|---|
| CSparse | 0.3601 | 20.7 ms | 0.66 | 15.6 ms | 0.67 | 9.6 ms | 0.72 | 6.6 | 0.76 |
| Sirius | 0.7127 | 24.1 ms | 0.78 | 18.2 ms | 0.78 | 11.1 ms | 0.83 | 7.7 ms | 0.88 |
| Full | 0.7612 | 30.9 ms | 1.0 | 23.2 ms | 1.0 | 13.3 ms | 1.0 | 8.6 ms | 1.0 |

Here we show that Sirius delivers its promised efficiency claim for on-chip and offloading settings. Because the fine-grained sparsity Lee et al. (2024) relies on a custom CUDA kernel to achieve the target generation speedup not open-sourced, we focus on coarse-grained sparsity on GSM-8K COT, and the input sequence length with average prefill length 900.

Firstly, we consider the on-chip setting running Llama-3-8B-Instruct on a single GPU. The sparse model (APU 0.65) achieves 36.01% accuracy on GSM8K-COT, while the full model achieves 76.12% accuracy on GSM8K-COT. With kernel size 10, SIRIUS achieves 0.74 APU with accuracy 71.27% accuracy. We use torch compile to optimize the inference latency and limit the overhead other than running model in-

Table 4: Llama-3-70B-Instruct with Offloading.

| Settings | Sparse | Sirius | Full |
|---|---|---|---|
| Performance | 0.7407 | 0.8719 | 0.9014 |
| Latency (s) | 3.57 s | 3.68 s | 5.72 s |
| Ratio to Full | 0.6241 | 0.6434 | 1.0 |

ference. The average latency generated per token is used to compute latency. Results are shown in Table 3. On average, SIRIUS delivers the promised latency reduction from APU calculations. The speedup ratio on A40 and L40 closely aligns with the theoretical APU reported. On the other hand, A100 and H100 compute MLP more efficiently than it compute attention, making the latency ratio between computing MLP and attention not perfectly aligned with their ratio in parameter size. Therefore, we see that even the sparse model baseline has slightly higher latency as expected. We increase the kernel size from 10 to 16 for these two devices, where accuracy reaches 0.7089 and the AAL reaches 13.67. For A100 and H100, building a hardware-efficient tree is nearly free of cost and highly effective. For numbers in Table 3, we use the width 4 tree that boosts the AAL to 15.01 out of 16. More details are in Appendix B.2.

Secondly, we consider the offloading setting which is the only way for resource-limited users to run 70B models by loading only the weights in use to GPU memory, while the others are offloaded to the CPU. Results are shown in Table 4. We use a single L40 48GB with a PCIe bus bandwidth of 25 GB/s to run Llama-3-70B-Instruct with batch size 1. Llama-3-70B-Instruct has roughly 80% of parameters to be MLP, which gives the theoretical APU for Griffin to be 0.6. Sparse + Sirius gives 0.649 APU, which is roughly what our system achieved.

## 6  Conclusion

We observe that contextual sparse methods significantly degrade reasoning and deduction tasks. SIRIUS, an efficient correction mechanism, enables accurate LLM inference with contextual sparsity. With roughly 11% to 18% sparsity increase, SIRIUS improves fine-grained and coarse-grained sparsity significantly in their performance while maintaining their efficiency gain.

Further, SIRIUS is still relying on rollback to correct the tokens that are deemed unlikely, which is inefficient. On the other hand, making the weak-strong model synergy systems that match the performance of the strong while keeping the efficiency of the weak, without strictly matching the strong models' output distribution remains an interesting and unsolved problem. We leave these topics to future works.

## 7  Acknowledgement

We would like to thank Feng Liang and Yunong Liu for their helpful feedback during the exploration and writing. We also want to give a special thanks to Hanshi Sun for providing insights and suggestions for efficient implementation and speedup.

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

# Appendix

## Appendix Table of Contents

## A  Additional Background

### A.1  Extended Related Works

**Pruning in LLM** Sparsity in neural networks has been widely studied. In the context of LLM, sparsity is studied under two branches - unstructured and structured. On the unstructured sparsity side, Frantar and Alistarh (2023) (SparseGPT) is a ground-breaking work that formulates pruning as a solving a series of sparse regression problems and proposes a fine solver for the problem. Sun et al. (2024b) (Wanda) introduces input activations into the pruning decision and achieves strong results in inducing LLM sparsity. On the structured side, LLMPruner Ma et al. (2023) and Sheared Llama Xia et al. (2024) each proposes different meticulous pruning algorithms and restoring weights through either parameter-efficient finetuning or efficient full weights training.

**Contextual Sparsity** Many recent works on LLM sparsity notice that the sparse pattern is highly related to the input or context. Deja Vu Liu et al. (2023b) revealed that for OPT models Zhang et al. (2022) the contextual sparsity is as high as 85%, meaning that 80% of the parameters can be pruned that won't hurt the token decoded quality given the prompt. Deja Vu formulates the problem of neuron selection as a near-neighbor search problem: finding neurons that are the most similar to the input activations. PowerInfer Song et al. (2023) extends the contextual sparsity to benefit the heterogeneous setting. Compared to the rest of the model, MLP layers tend to possess significant contextual sparsity and can be effectively exploited in a training-free manner. Concurrently, Griffin Dong et al. (2024) discovers the phenomenon of flocking, where MLP neurons have temporal locality, where given a fixed prompt, similar neurons tend to get activated throughout the following generation. Flocking is shown to occur in most activation types and open-source LLMs. Griffin selects the same set of heated neurons with 50% sparsity throughout the generation of each input prompt, which we refer to as coarse-grained sparsity. CATS Lee et al. (2024) successfully exploits per-token contextual sparsity in the MLP layers for inference latency reduction. They resample a new set of neurons per every new input token, which we categorize it as fine-grained contextual sparsity. Our paper mainly focuses on the training-free MLP sparsity techniques. Although these recent works show minimal accuracy degradation in classification and easy text summarization tasks, they both severely degrade in generation quality under tasks that require high-level reasoning and understanding ability. Our work serves as a low-cost complementary tool, aiming to push these elegant and promising techniques for mainstream use cases.

Also, previous contextual sparsity methods haven't fully and exhaustively evaluated their benefits and limitations in downstream generation tasks. To fully study this technique, we extensively go through

open-source LLMs in diverse performance and sizes on diverse generation tasks and datasets to locate where these sparse models maintain the performance or fail.

**Speculative Decoding** Besides model compression techniques, Speculative decoding Leviathan et al. (2023), Chen et al. (2023), Kim et al. (2023) is another important LLM inference latency reduction method. Compared to LLM, small transformer models are much more computationally accessible and can effectively model short-range tokens. Therefore, smaller models are asked to speculate short-term future tokens, which the LLM takes in in parallel to trade in FLOPs with memory loading time. During verification, most speculative decoding methods pursue lossless acceleration, leading to frequent rollback during rejection. In contrast, Sirius solves a very different problem. Our method aims to maximally preserve the efficiency of sparse models while boosting its performance. Sparse models, pruned directly from LLM, are much stronger at modeling a longer range of text than draft models, thus requiring much less help from the LLM. Our work aims to find the minimum amount of LLM overhead while boosting its performance to the LLM level. Given the resemblance and relevance of Speculative Decoding to our method Sirius, we will elaborate more in-depth on their differences and Speculative Decoding's inefficiencies when it comes to helping the Sparse method in A.2.

### A.2 Why Not Using the Speculative Decoding to Correct the Sparse Model?

When Speculative Decoding is used to correct sparse using the full model, we will show that the efficiency of the overall process will be largely limited. We followed the common practice from speculative decoding and measured the acceptance rate on different datasets C4 Raffel et al. (2020) and GSM8K Cobbe et al. (2021). Take the Coarse-grained sparse model as an example. For Llama-3-8B as the full model, the 50% sparse (APU 0.65) model will produce an acceptance rate of 0.71 on C4 and 0.89 on GSM8K. Speculative decoding also use parallel verification in the period-basis. Naturally, to keep the system efficiency high, we need to (1) enlarge the period and (2) increase the average number of tokens accepted (AAL) given the gamma (period - 1) value. Take the acceptance rate of 0.89 on GSM8K as an example, following the formulation in Leviathan

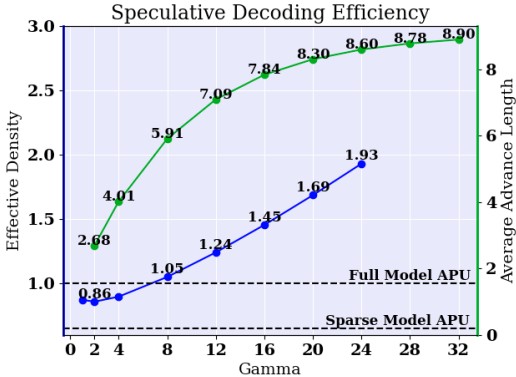

Figure 6: Speculative Decoding has limitation in efficiency when correcting sparse models.

et al. (2023), we can calculate the expected number of accepted tokens for every gamma term in the Speculative Decoding literature. $AAL = \frac{1-\alpha^{(\gamma+1)}}{1-\alpha}$. The trend (green) is plotted in Figure 6

We can notice the trend that the average advance length starts to plateau as the gamma becomes larger. Take the gamma of 16 as an example, the period is then 17. The average advance length is only 7.84. The APU is (16 * 0.65 + 1)/7.84 = 1.45, which is larger than the full model 1. The blue line in Figure 6 shows the relationship between APU and gamma.

Because of the plateauing effect, for an acceptance rate of 0.89, the best gamma is 2 (period = 3). The optimal APU is 0.86, compared with 0.65 coarse-grained sparse APU. A similar picture can be applied to Fine-grained sparsity as well. The key reasons for the observation are two-fold: (1) the contextually sparse models are too big to be the draft model of the speculative decoding system and to have a large period; (2) Speculative decoding preserves the original model's performance so that the acceptance criteria are usually very strict, which is also not suitable for large period and high average advance length. Following the same spirit, Sun et al. (2024a) also uses a large draft model to do self-speculation, but for them, the authors select gamma = 1 to achieve the optimal speedup of their system. **In contrast, SIRIUS brings <0.76 APU in this case with period $\geq$ 10.** Specifically, with a threshold of 0.1, Sirius can correct Llama-3-8B coarse-grained sparsity from 20.85% to 43.9%, compared to the 49.66% full model. With a period of 16 tokens (gamma = 15), Sirius on average can accept 13.4 tokens out of a kernel size of 16 and over 9 tokens out of a kernel size of 10, translating to APU < 0.76, significantly lower than SD does.

# B Supplemental Details in SIRIUS observations, Design, and Experiments

In this section, we provide several supplemental experiments to the picture. First, we run SIRIUS on Llama-3-70B. However, because of computational limits, we cannot run SIRIUS with the tree on Llama-3-70B with the scale we did for other models. Nevertheless, we do show that 70B has roughly the same pattern as we have seen before, large model sparsity also somehow struggles on reasoning tasks. Second, we provide additional proof for the parallel verification efficiency statement. After that, I show results on where the error is located in the chunk size of 16 tokens. The error is distributed almost uniformly. Last but not least, we also apply SIRIUS on datasets that are reasoning. Lastly, we provide more results on the comparison between models of similar size but have a huge performance gap. We show that given the similar parameter size, the trend is for a more well-trained, powerful model to degrade more from contextual sparsity. We present here more illustration on contextual sparsity and Llama-2-7B-Chat experiments with fixing only a minor portion of tokens is shown in Figure 7.

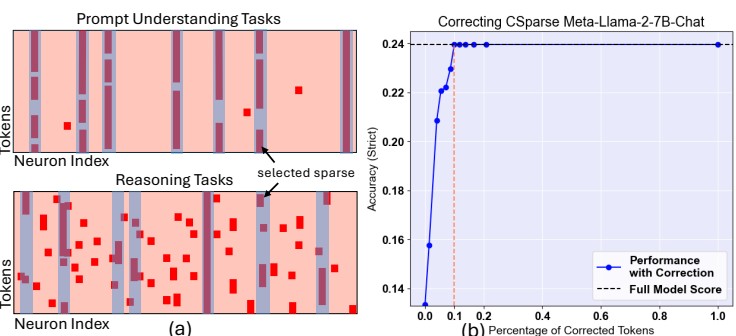

Figure 7: (a) Illustration on why Contextual Sparsity has uneven performance on different tasks. The activation heat map (red) has the brighter the color the larger in magnitude. On top, we also show the neuron sparsity selected. The graph points signify that the pattern in the prompt understanding task is easier to capture. (b) An additional graph of correcting Csparse Llama-2-7B-Chat. It is similar to the previous experiment on 8B. Only 10% tokens being corrected results in complete performance recovery.

## B.1 Given Similar Parameter Size Well-trained Models Suffer More

We observe another interesting phenomenon: given the similar parameter size, the more well-trained the model is, the more performance degradation contextual sparsity would make on the full models. Here we present two pairs of results. First, we look at the performance between Llama-3-8B-Instruct and Llama-2-7B-Chat with Llama-3-70B-Instruct and Llama-2-70B-Chat. All models are evaluated on GSM8K-COT. We draw these models in CSparse in Figure 8, and the readers can find more results in Appendix B.8. We can see figures from top to bottom, where even at lower density (more elements are not selected), Llama-2-7B-Chat and Llama-2-70B-Chat suffer from less performance degradation (blue) compared to the Llama-3-8B-Instruct and Llama-3-70B-Instruct models. Furthermore, suppose we focus on Llama-3-70B-Instruct for global density at 60% or lower. In that case, the performance (coral) is degraded significantly, which is comparable or even lower to Llama-3-8B-Instruct full model performance at 0.76, Even at 50% density, the 70B model still has more than 40B parameters, much more expensive than the 8B model. The observation fully manifests the difficulty of using CS in complex reasoning tasks.

## B.2 Hardware Friendly Tree Building Process

In this section, we first look at the insights behind whether building the tree can help efficiency, then we detail the specific steps towards tree pruning.

The goal for the Sirius system is to make $n_{AAL}$ to be as large as possible. Despite the full model sharing KVs with the sparse model, Sirius still encounters costly rollbacks because of sparse greedily decoded tokens being rejected. Interestingly, we look closely into where the sparse model is likely to make a mistake

Table 5: The second and third most likely tokens from sparse models offer potential for boosting efficiency.

| Sparsity | $2^{nd}$ **Hit** | $3^{rd}$ **Hit** | **Miss** | **Coverage%** |
|---|---|---|---|---|
| **FSparse** | 79% | 11% | 9% | 90% |
| **CSparse** | 65% | 17% | 16% | 82% |

on GSM8K and AQuA-RAT-COT Ling et al. (2017) with Sirius on Llama-3-8B-Instruct and a kernel size of 16. More details are shown in Appendix B.6. The error distributes almost uniformly across all positions of the kernel size. Also, when the token makes the mistake, besides the greedily decoded

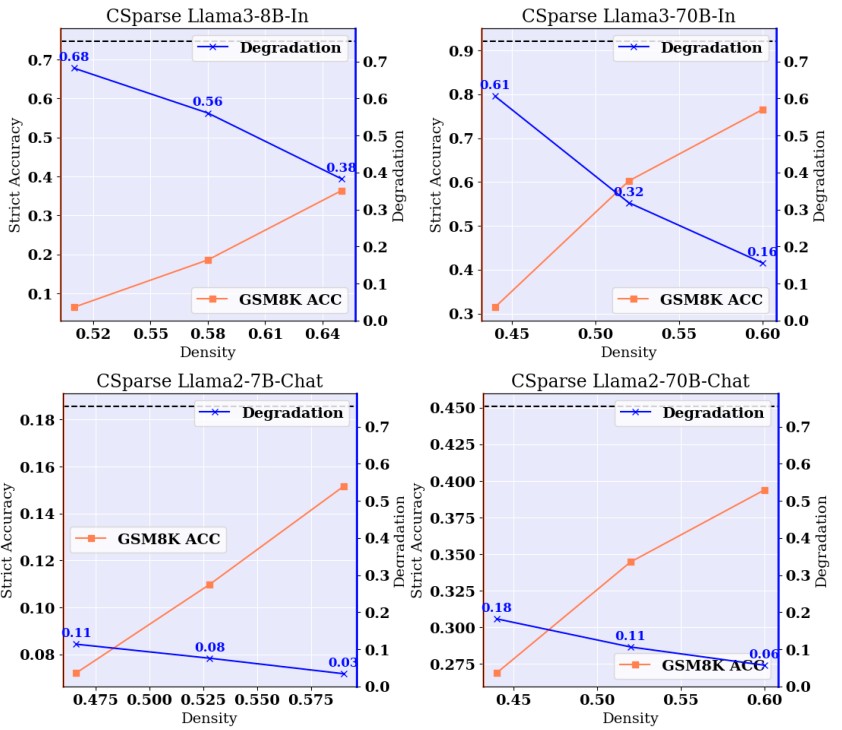

Figure 8: Given the similar model parameters, the more well-trained the model is, the worse the degradation would be. (Compare the figures vertically between Llama-3 and Llama-2 family models).

tokens, we find that other tokens of lower likelihood offer the potential to boost efficiency. Surprisingly, we found that out of the cases where the greedily decoded tokens are rejected, the probability that the second or third most likely tokens from the sparse being accepted by the full model is reasonably high.

Shown in Table 5, we test on part of the GSM8K rejected cases. The "Second Hit" is defined as the count of the second most likely tokens being accepted by the full model when the greedily decoded token is rejected, while the "Third Hit" is defined as the count of the third most likely token being accepted when the first two are rejected. Both sparsity method has a high acceptance rate, or "Coverage", from the second and third most likely tokens when the most likely token is rejected, showing huge potential for gains in efficiency.

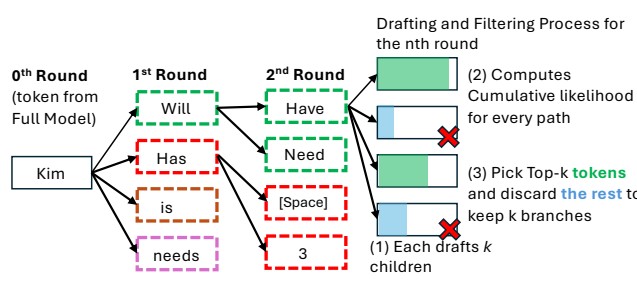

Figure 9: Illustration of Tree Building Process.

To capitalize the potential from the second to third tokens, we propose to build a tree during the sparse generating process (lines 6 to 11 in Algorithm 1. The tree algorithm is similar to Beam Search Freitag and Al-Onaizan (2017). However, to make sure that the tree building and tree parallel correction processes can achieve speedup over cases that don't build trees, we impose strong restrictions on the tree structure we build. For a fixed kernel size, we limit every step to having a fixed number of leaves, or treewidth, through tree pruning based on ranking the cumulative log-likelihood of the path. The resulting tree has a fixed shape for a given kernel size and tree width, but only the interconnection pattern between steps varies based on the pruning and ranking within each step. The details are illustrated in Figure 9. During verification, out of the treewidth complete paths, we select the one that reaches the longest advance length. In practice, we found that for kernel size 16, when the treewidth is increased to

8, the optimal verification tree is around 64. From Section A.2, we see that the parallel verification of the tree of 64 roughly equals the time the full input 1 token.

Therefore, a treewidth of 8 is set as the maximum treewidth when building the tree for kernel size 16 for later. We show that building a tree makes the system significantly more efficient while retaining correction effects.

Table 6: Performance and Speedup Ratios on GSM8K-COT with tree building, latency measurement in millisecond.

| Settings | ACC | A100 | Ratio | H100 | Ratio |
|---|---|---|---|---|---|
| **CSparse** | 0.3601 | 9.6 | 0.72 | 6.6 | 0.76 |
| **Sirius** (No Tree) | 0.7127 | 11.8 | 0.88 | 8.2 | 0.95 |
| **Sirius** (With Tree) | 0.7089 | 11.1 | 0.83 | 7.7 | 0.88 |
| **Full** | 0.7612 | 13.3 | 1.0 | 8.6 | 1.0 |

In Table 6, we present more details complementing Table 3 on the use of tree-building for A100 and H100. We ablate the use of a width 4 tree by not using the tree under kernel size 16. We observe that the speedup improvements solely from the increase in AAL from 13.67 to 15.01 for kernel size 16.

## B.3 Ablation: Various Aspects of Sirius Are Tested and Challenged

Table 7: Ablation on Components in Sirius.

| CSparse | GSM8K 20% | FSparse | GSM8K 20% |
|---|---|---|---|
| **Llama3-8B-Instruct** | 0.7538/0.7538 | **Llama3-8B-Instruct** | 0.7538/0.7538 |
| + CSparse | 0.3674/0.3674 | + FSparse | 0.5644/0.5644 |
| + CSparse + Interleave | 0.3826/0.3826 | + FSparse + Interleave | 0.6288/0.6288 |
| + CSparse + KV Rewrite | 0.4735/0.4735 | + FSparse + KV Rewrite | 0.6629/0.6629 |
| + CSparse + KV Rewrite + Interleave | 0.4886/0.4886 | + FSparse + KV Rewrite + Interleave | 0.6780/0.6818 |
| + CSparse + Roll back + Interleave | 0.6591/0.6591 | + FSparse + Roll back + Interleave | 0.7273/0.7273 |
| + CSparse + KV Rewrite + Interleave + Rollback | 0.6667/0.6667 | + FSparse + KV Rewrite + Interleave + Rollback | 0.7273/0.7311 |

**Probing Components** To understand the contribution and the utility of each component of Sirius, we ablate all components of Sirius in Table 7. We started by only letting the LLM correct the token it is evaluating (interleaving only). Then, we add on top of it the KV cache correction, and then the rollback. All these three techniques are effective when applied solely. Rollback seems to be the most effective technique. Even when applied alone, rollback asserts significant correction to both the CSparse and FSparse models. Interestingly, KV Cache is also effective alone, bringing a 12% increment for CSparse and an 11% accuracy increase for FSparse. Relatively, interleaving is the weakest. Surprisingly, adding both KV rewriting and rollback is only marginally better than rollback alone. Although it is tempting to think KV Cache rewriting is not useful with rollback, the improvement KV Cache Rewriting brings is a gain in efficiency. When adding the KV Cache Rewriting on top of Roll Back and interleave it significantly improves the efficiency of the correction. For CSparse, adding KV rewrite increases AAL from 12.77 to 13.80.

Table 8: Ablation on the threshold for correction (FSparse Llama-3-8B-Instruct).

| Threshold | Full | Sparse | 0.05 | 0.1 | 0.3 | 0.5 | 0.7 | 0.9 |
|---|---|---|---|---|---|---|---|---|
| **Accuracy** | 0.7803 | 0.5884 | 0.7247 | 0.7399 | 0.7399 | 0.7677 | 0.7702 | 0.7652 |
| **AAL** | N/A | N/A | 15.2 | 14.6 | 11.6 | 8.5 | 6.2 | 4.2 |

**Likelihood threshold to balance Correction and Efficiency** We found that the likelihood threshold is important for managing the Sirius correction and efficiency tradeoff. We present results in Table 8. We ablate this setting on a 30% subsampled GSM8K dataset, and only strict accuracy is reported. The

performance is the score, while the efficiency is measured by Average Advance Length (AAL). We can find that with the increase of threshold, the scores generally improve, while the efficiency metric decreases.

**Building Wider Tree** We study the effect of increasing the treewidth. In fact, for every number from SIRIUS in Table 2, we are selecting from a group of results by different treewidth. We present all of this treewidth and its corresponding accuracy and efficiency numbers in the Appendix C. *Importantly, raising treewidth always improves AAL.* Although different choices of treewidth usually give similar accuracy scores, there is hardly a pattern on which treewidth always gives the best accuracy. The optimal treewidth can only be found through empirical studies.

## B.4 Large Model Experiments

To diversify the evaluation of Sirius, we also evaluate Sirius's Effectiveness on the Llama-3-70B-Instruct model. MMLU is subsampled 10%, while CNN/DailyMail is subsampled 30%. The following table contrasts with Llama-3-8B-Instruct. We use strict match/flexible extract accuracy for GSM-8K-COT, accuracy for MMLU, F1/EM score for CoQA, Rouge-1/2/L score for CNN/DailyMail, and Rouge-1/2 ACC for TruthfulQA.

Table 9: Large model results on miscellaneous datasets.

| | GSM-8K-COT | MMLU | CoQA | CNN/DailyMail | TruthfulQA |
|---|---|---|---|---|---|
| **Llama-3-70B-In** | 0.9014/0.9022 | 0.7456 | 0.6567/0.8069 | 0.1016/0.0206/0.0964 | 0.5116/0.4247 |
| **+ CSparse** | 0.7407/0.7483 | 0.7018 | 0.6497/0.8046 | 0.1019/0.0208/0.0967 | 0.4541/0.3807 |
| **+ FSparse** | 0.8726/0.8772 | 0.7193 | 0.6497/0.8035 | 0.1015/0.0206/0.0963 | 0.4835/0.3905 |
| **Llama-3-8B-In** | 0.7612/0.7672 | 0.6272 | 0.6153/0.7825 | 0.1015/0.0204/0.0963 | 0.4945/0.3647 |
| **+ CSparse** | 0.3601/0.3647 | 0.5307 | 0.6003/0.7735 | 0.1016/0.0206/0.0964 | 0.5067/0.3953 |
| **+ FSparse** | 0.6103/0.6202 | 0.4825 | 0.5828/0.7577 | 0.1017/0.0204/0.0965 | 0.5202/0.3941 |

## B.5 Variable Sequence Length with Batch Size One

Table 10: A100 Latency versus Input Sequence Length.

| Input Sequence Length | A100 Latency (ms) |
|---|---|
| 1 | 0.0133 |
| 2 | 0.0135 |
| 4 | 0.0136 |
| 8 | 0.0138 |
| 16 | 0.0140 |
| 32 | 0.0149 |
| 64 | 0.0144 |
| 96 | 0.0171 |

Here we show the benchmark latency on A100, where the input tensor to Llama-3-8B-Instruct has a shape of batch size 1 and a different input sequence length. To get the hardware optimal readings, we use torch compile to compile the whole forward pass of the model. We show that the latency only goes up insignificantly to 64, but the trend of increment to 96 is a bit steep.

## B.6 Error Occurs At Which Position inside a Chunk

We look at the distribution of where the error would be inside a kernel of 16 tokens. We run through Sirius with a kernel size of 16 on the entire GSM-8K and AQuA-RAT-COT dataset. The histogram is shown in Figure 10. We found that the error occurs in a uniform pattern, where it is hard to see any particular region where the tokens are likely to occur the most.

## B.7 Miscellaneous Results

Besides, the results on the complex reasoning tasks, we evaluate Sirius on slightly more diverse tasks in Table 12.

Figure 10: We look at the histogram of the number of errors versus the position among a period of sixteen tokens on average. We have two different datasets of Arithmetic Reasoning GSM-8K and AQuA-RAT-COT. We can see that the number of errors is distributed almost evenly for both datasets.

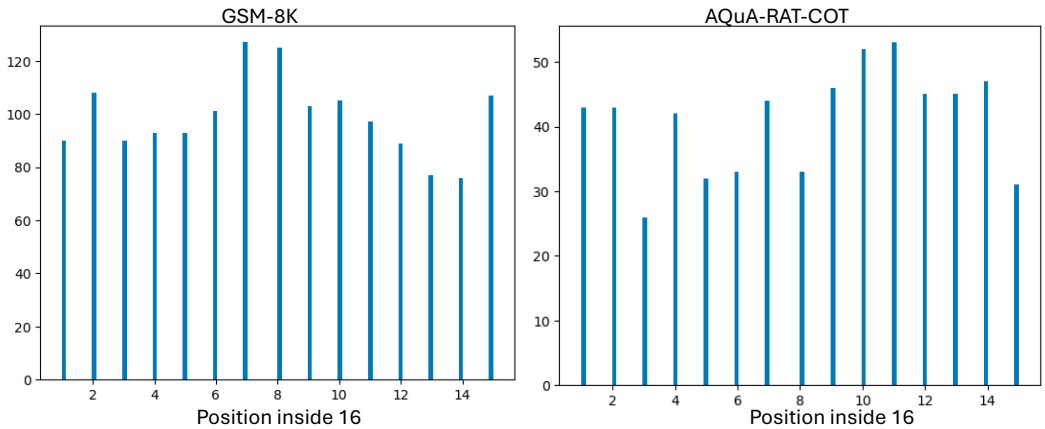

## B.8   Llama-2 and Llama-3 Models on GSM8K-COT

Table 11: Detail on Llama-2 and Llama-3 family models with CS.

| Llama-3-70B-Instruct | Accuracy | Degradation | Llama-3-8B-Instruct | Accuracy | Degradation |
|---|---|---|---|---|---|
| Full | 0.9205 | | Full | 0.7462 | |
| Csparse 60% | 0.8144 | 0.1061 | | | |
| Csparse 50% | 0.7652 | 0.1553 | Csparse 50% | 0.3636 | 0.3826 |
| Csparse 40% | 0.6023 | 0.3182 | Csparse 40% | 0.1856 | 0.5606 |
| Csparse 30% | 0.3144 | 0.6061 | Csparse 30% | 0.0644 | 0.6818 |
| Fsparse 50% | 0.8864 | 0.0341 | Fsparse 50% | 0.6477 | 0.0985 |
| Fsparse 40% | 0.8485 | 0.0720 | Fsparse 40% | 0.4053 | 0.3409 |
| Fsparse 30% | 0.7386 | 0.1819 | Fsparse 30% | 0.0265 | 0.7197 |
| Fsparse 20% | 0.2803 | 0.6402 | | | |
| **Llama-2-70B-Chat** | **Accuracy** | **Degradation** | **Llama-2-7B-Chat** | **Accuracy** | **Degradation** |
| Full | 0.4508 | | Full | 0.1856 | |
| Csparse 50% | 0.3939 | 0.0569 | Csparse 50% | 0.1515 | 0.0341 |
| Csparse 40% | 0.3447 | 0.1061 | Csparse 40% | 0.1098 | 0.0758 |
| Csparse 30% | 0.2689 | 0.1819 | Csparse 30% | 0.0720 | 0.1136 |
| Fsparse 50% | 0.3864 | 0.0644 | Fsparse 50% | 0.1629 | 0.0227 |
| Fsparse 40% | 0.3902 | 0.0606 | Fsparse 40% | 0.1364 | 0.0492 |
| Fsparse 30% | 0.2689 | 0.1819 | Fsparse 30% | 0.1212 | 0.0644 |

Here we present more experiments for the comparison between Llama-2 and Llama-3 family models, which is first mentioned in Section B.1, where we also include FSparse methods together with the CSparse method. The results are in Table 11.

Table 12: Miscellaneous Results: 5 models on different Three Different datasets.

| Experiment Setting | CoQA | AGIEval (Math) | MMLU-FLAN-COT |
|---|---|---|---|
| Llama-2-7B-Chat | 0.5982/0.7580 | 0.072 | 0.4925 |
| Llama-2-7B-Chat-FSparse | 0.5898/0.7540 | 0.077 | 0.4768 |
| Llama-2-7B-Chat-FSparse-Sirius | 0.5908/0.7540 | 0.081 | 0.4670 |
| Llama-2-7B-Chat-CSparse | 0.6117/0.7639 | 0.065 | 0.4637 |
| Llama-2-7B-Chat-CSparse-Sirius | 0.6117/0.7664 | 0.078 | 0.4794 |
| Llama-3-8B-Instruct | 0.6153/0.7825 | 0.213 | 0.6231 |
| Llama-3-8B-Instruct-FSparse | 0.5828/0.7577 | 0.172 | 0.5304 |
| Llama-3-8B-Instruct-FSparse-Sirius | 0.5868/0.7591 | 0.196 | 0.5709 |
| Llama-3-8B-Instruct-CSparse | 0.6003/0.7735 | 0.154 | 0.5558 |
| Llama-3-8B-Instruct-CSparse-Sirius | 0.6005/0.7728 | 0.178 | 0.6003 |
| Llama-2-13B-Chat | 0.6408/0.7896 | 0.092 | 0.5317 |
| Llama-2-13B-Chat-FSparse | 0.6320/0.7837 | 0.087 | 0.5082 |
| Llama-2-13B-Chat-FSparse-Sirius | 0.6340/0.7859 | 0.089 | 0.5219 |
| Llama-2-13B-Chat-CSparse | 0.6350/0.7841 | 0.088 | 0.5127 |
| Llama-2-13B-Chat-CSparse-Sirius | 0.6363/0.7847 | 0.1 | 0.5127 |
| Llama-2-7B | 0.6388/0.7735 | 0.101 | 0.4520 |
| Llama-2-7B-FSparse | 0.6352/0.7697 | 0.09 | 0.4435 |
| Llama-2-7B-FSparse-Sirius | 0.6352/0.7697 | 0.092 | 0.4415 |
| Llama-2-7B-CSparse | 0.6338/0.7700 | 0.086 | 0.4213 |
| Llama-2-7B-CSparse-Sirius | 0.6372/0.7709 | 0.093 | 0.4317 |
| Llama-3-8B | 0.6727/0.8055 | 0.163 | 0.5754 |
| Llama-3-8B-FSparse | 0.6625/0.7984 | 0.152 | 0.5349 |
| Llama-3-8B-FSparse-Sirius | 0.6625/0.7984 | 0.154 | 0.5532 |
| Llama-3-8B-CSparse | 0.6633/0.7977 | 0.131 | 0.5049 |
| Llama-3-8B-CSparse-Sirius | 0.6670/0.7995 | 0.15 | 0.5428 |

## C  Additional Results on Reasoning

Due to page restrictions, we only show GSM8K, CSQA, and HumanEval in the paper. Below we show additional results to the numbers presented in the paper. **We present tables of a similar format. Please notice that the leftmost column writes a number that represents the treewidth in the given settings.** Also, we show the results of Sirius on the other five datasets AQuA-RAT-COT (Arithmetic Reasoning), Sports (Commonsense Reasoning), Date (Commonsense Reasoning), and StrategyQA (CommonSense Reasoning), and MBPP+ (coding).

### C.1  Arithmetic Reasoning

In this section, we present GSM8K and AQuA RAT COT evaluation results with the efficiency metric AAL. Sirius is shown to be effective on these two reasoning tasks about arithmetic. Below we show the raw AAL score associated with efficiency for all models and the performance of different treewidths.

### C.2  CommonSense Reasoning

We followed the COT paper and evaluated Sirius on CSQA, Sports, StrategyQA, and Dates. Sparse methods are capable of outputting high-quality output similar to the full model at the 0.5 mark, which is different than on other datasets. However, we tune the sparsity level to 0.4 (0.6 dense, 0.4 removed), and it starts to have performance degradation. Sirius can compensate them with relatively high efficiency)

### C.3  Code

We also have a coding portion that evaluates Sirius on HumanEval. Sirius performs well similar to other datasets. Besides, we also have results on MBPP+. The results show Sirius effectiveness and efficiency again.

Table 13: SIRIUS Tree on GSM8K.

| Experiment Settings treewidth | Llama-3-8B-Instruct-FSparse | | Llama-3-8B-Instruct-CSparse | |
|---|---|---|---|---|
| | Performance | AAL (out of 16) | Performance | AAL (out of 16) |
| Original Performance | 0.7536/0.7544 | N/A | 0.7536/0.7544 | N/A |
| Sparse Performance | 0.5868/0.5891 | N/A | 0.3844/0.3867 | N/A |
| 1 | 0.7316/0.7324 | 14.5903 | 0.6983/0.7005 | 13.1903 |
| 2 | 0.7172/0.7172 | 14.9554 | 0.7089/0.7096 | 14.1517 |
| 4 | 0.7278/0.7309 | 15.3705 | 0.7119/0.7111 | 14.8393 |
| 6 | 0.7195/0.7187 | 15.5979 | 0.7081/0.7074 | 15.0682 |
| 8 | 0.7202/0.7218 | 15.5548 | 0.7051/0.7058 | 15.2291 |
| | Llama-3-8B-FSparse | | Llama-3-8B-CSparse | |
| | Performance | AAL (out of 16) | Performance | AAL (out of 16) |
| Original Performance | 0.4966/0.5042 | N/A | 0.4966/0.5042 | N/A |
| Sparse Performance | 0.3199/0.3260 | N/A | 0.2085/0.2168 | N/A |
| 1 | 0.4526/0.4572 | 14.6946 | 0.439/0.445 | 13.361 |
| 2 | 0.4579/0.4640 | 15.0355 | 0.4299/0.4367 | 14.3061 |
| 4 | 0.4579/0.4540 | 15.0355 | 0.4223/0.4306 | 14.9721 |
| 6 | 0.4450/0.4503 | 15.4834 | 0.4177/0.4238 | 15.1435 |
| 8 | 0.4352/0.4428 | 15.5863 | 0.4177/0.4238 | 15.2939 |
| | Llama-2-13B-Chat-FSparse | | Llama-2-13B-Chat-CSparse | |
| | Performance | AAL (out of 16) | Performance | AAL (out of 16) |
| Original Performance | 0.3548/0.3647 | N/A | 0.3548/0.3647 | N/A |
| Sparse Performance | 0.3222/0.3275 | N/A | 0.2714/0.2767 | N/A |
| 1 | 0.3533/0.3472 | 15.085 | 0.3412/0.3472 | 14.0153 |
| 4 | 0.3412/0.3374 | 15.7576 | 0.3381/0.3412 | 15.3491 |
| | Llama-2-13B-FSparse | | Llama-2-13B-CSparse | |
| | Performance | AAL (out of 16) | Performance | AAL (out of 16) |
| Original Performance | 0.2282/0.2312 | N/A | 0.2282/0.2312 | N/A |
| Sparse Performance | 0.2191/0.2229 | N/A | 0.1759/0.1797 | N/A |
| 1 | 0.2328/0.2381 | 15.6759 | 0.2418/0.2472 | 15.3415 |
| 4 | 0.2372/0.2403 | 15.9283 | 0.2077/0.2100 | 15.825 |
| | Llama-2-7B-Chat-FSparse | | Llama-2-7B-Chat-CSparse | |
| | Performance | AAL (out of 16) | Performance | AAL (out of 16) |
| Original Performance | 0.2403/0.2426 | N/A | 0.2403/0.2426 | N/A |
| Sparse Performance | 0.1971/0.1994 | N/A | 0.1334/0.1372 | N/A |
| 1 | 0.2282/0.2312 | 14.8172 | 0.2214/0.2229 | 12.6888 |
| 2 | 0.2297/0.2305 | 15.1784 | 0.2252/0.2359 | 13.8875 |
| 4 | 0.2305/0.2282 | 15.5467 | 0.2183/0.2214 | 14.6751 |
| 6 | 0.2388/0.2411 | 15.691 | 0.2199/0.2206 | 14.8575 |
| 8 | 0.2312/0.2343 | 15.735 | 0.2244/0.2252 | 15.0017 |
| | Llama-2-7B-FSparse | | Llama-2-7B-CSparse | |
| | Performance | AAL (out of 16) | Performance | AAL (out of 16) |
| Original Performance | 0.1357/0.1403 | N/A | 0.1357/0.1403 | N/A |
| Sparse Performance | 0.1137/0.1168 | N/A | 0.0758/0.0804 | N/A |
| 1 | 0.1183/0.1205 | 15.6864 | 0.1152/0.1168 | 15.1096 |
| 2 | 0.1334/0.1357 | 15.7893 | 0.113/0.116 | 15.5358 |
| 4 | 0.1410/0.1448 | 15.9161 | 0.113/0.116 | 15.8558 (53.341) |
| 6 | 0.1289/0.1312 | 15.9662 | 0.1183/0.1205 | 15.8715 |
| 8 | 0.1236/0.1259 | 15.9568 | 0.1114/0.1145 | 15.8939 |

Table 14: SIRIUS on AQuA-RAT-COT.

| Experiment Settings treewidth | Llama-3-8B-Instruct-FSparse | | Llama-3-8B-Instruct-CSparse | |
|---|---|---|---|---|
| | Performance | AAL (out of 16) | Performance | AAL (out of 12) |
| Original Performance | 0.515748 | N/A | 0.515748 | N/A |
| Sparse Performance | 0.42126 | N/A | 0.271654 | N/A |
| 1 | 0.429134 | 13.1945 | 0.468504 | 10.3322 |
| 4 | 0.488189 | 14.4722 | 0.44095 | 10.6228 |
| 6 | 0.496063 | 15.115 | 0.452756 | 10.8348 |
| 8 | 0.476378 | 15.3721 | 0.456693 | 11.0874 |
| | Llama-3-8B-FSparse | | Llama-3-8B-CSparse | |
| | Performance | AAL (out of 16) | Performance | AAL (out of 12) |
| Original Performance | 0.456693 | N/A | 0.456693 | N/A |
| Sparse Performance | 0.287402 | N/A | 0.228346 | N/A |
| 1 | 0.377953 | 12.6665 | 0.377945 | 12.6665 |
| 4 | 0.385827 | 14.46 | 0.397638 | 10.2826 |
| 6 | 0.366142 | 15.0671 | 0.370079 | 10.6753 |
| 8 | 0.42126 | 15.0995 | 0.38189 | 11.0019 |
| | Llama-2-13B-Chat-FSparse | | Llama-2-13B-Chat-CSparse | |
| | Performance | AAL (out of 16) | Performance | AAL (out of 12) |
| Original Performance | 0.232283 | N/A | 0.232283 | N/A |
| Sparse Performance | 0.251969 | N/A | 0.208661 | N/A |
| 1 | 0.275591 | 15.5163 | 0.26378 | 9.17465 |
| 4 | 0.279528 | 15.3995 | 0.259843 | 10.8726 |
| | Llama-2-13B-FSparse | | Llama-2-13B-CSparse | |
| | Performance | AAL (out of 16) | Performance | AAL (out of 12) |
| Original Performance | 0.149606 | N/A | 0.149606 | N/A |
| Sparse Performance | 0.165354 | N/A | 0.149606 | N/A |
| 1 | 0.185039 | 15.4652 | 0.161417 | 11.0874 |
| 4 | 0.220472 | 15.6346 | 0.192913 | 11.8238 |
| | Llama-2-7B-Chat-FSparse | | Llama-2-7B-Chat-CSparse | |
| | Performance | AAL (out of 16) | Performance | AAL (out of 16) |
| Original Performance | 0.251969 | N/A | 0.251969 | N/A |
| Sparse Performance | 0.283465 | N/A | 0.220472 | N/A |
| 1 | 0.248031 | 15.5294 | 0.251969 | 12.6424 |
| 4 | 0.259843 | 15.7254 | 0.244096 | 14.5794 |
| | Llama-2-7B-FSparse | | Llama-2-7B-CSparse | |
| | Performance | AAL (out of 16) | Performance | AAL (out of 16) |
| Original Performance | 0.15748 | N/A | 0.15748 | N/A |
| Sparse Performance | 0.153543 | N/A | 0.177165 | N/A |
| 1 | 0.185039 | 15.423 | 0.141732 | 15.4122 |
| 4 | 0.161417 | 15.651 | 0.145669 | 15.3788 |

Table 15: SIRIUS on CSQA.

| Experiment Settings treewidth | Llama-3-8B-Instruct-FSparse | | Llama-3-8B-Instruct-CSparse | |
|---|---|---|---|---|
| | **Performance** | **AAL (out of 16)** | **Performance** | **AAL (out of 16)** |
| Original Performance | 0.707341 | N/A | 0.707341 | N/A |
| Sparse Performance | 0.615889 | N/A | 0.647011 | N/A |
| 1 | 0.699427 | 12.2108 | 0.724816 | 11.0512 |
| 4 | 0.687961 | 13.2734 | 0.709255 | 13.5876 |
| 6 | 0.714169 | 13.7842 | 0.720721 | 13.3097 |
| 8 | 0.710893 | 14.1173 | 0.707617 | 14.76893 |
| | **Llama-3-8B-FSparse** | | **Llama-3-8B-CSparse** | |
| | **Performance** | **AAL (out of 16)** | **Performance** | **AAL (out of 16)** |
| Original Performance | 0.643735 | N/A | 0.643735 | N/A |
| Sparse Performance | 0.53317 | N/A | 0.558559 | N/A |
| 1 | 0.638821 | 15.0088 | 0.618346 | 12.7426 |
| 4 | 0.630631 | 14.6151 | 0.63964 | 14.8704 |
| 6 | 0.625717 | 14.9905 | 0.640459 | 15.1968 |
| 8 | 0.617527 | 15.2534 | 0.642916 | 15.4355 |
| | **Llama-2-13B-Chat-FSparse** | | **Llama-2-13B-Chat-CSparse** | |
| | **Performance** | **AAL (out of 16)** | **Performance** | **AAL (out of 12)** |
| Original Performance | 0.687961 | N/A | 0.687961 | N/A |
| Sparse Performance | 0.53317 | N/A | 0.553645 | N/A |
| 1 | 0.657658 | 13.0868 | 0.649468 | 9.2183 |
| 4 | 0.669124 | 14.309 | 0.669124 | 11.438 |
| | **Llama-2-13B-FSparse** | | **Llama-2-13B-CSparse** | |
| | **Performance** | **AAL (out of 16)** | **Performance** | **AAL (out of 16)** |
| Original Performance | 0.610975 | N/A | 0.610975 | N/A |
| Sparse Performance | 0.570025 | N/A | 0.560197 | N/A |
| 1 | 0.578215 | 15.2554 | 0.58231 | 14.7381 |
| 4 | 0.586405 | 15.7213 | 0.606061 | 15.7284 |
| | **Llama-2-7B-Chat-FSparse** | | **Llama-2-7B-Chat-CSparse** | |
| | **Performance** | **AAL (out of 16)** | **Performance** | **AAL (out of 16)** |
| Original Performance | 0.624898 | N/A | 0.624898 | N/A |
| Sparse Performance | 0.616708 | N/A | 0.520066 | N/A |
| 1 | 0.632269 | 14.1015 | 0.608518 | 11.4607 |
| 4 | 0.638002 | 15.0978 | 0.605242 | 14.0366 |
| 6 | 0.605242 | 15.4365 | 0.611794 | 14.7197 |
| 8 | 0.621622 | 15.552 | 0.617527 | 15.0799 |
| | **Llama-2-7B-FSparse** | | **Llama-2-7B-CSparse** | |
| | **Performance** | **AAL (out of 16)** | **Performance** | **AAL (out of 16)** |
| Original Performance | 0.474201 | N/A | 0.474201 | N/A |
| Sparse Performance | 0.471744 | N/A | 0.441441 | N/A |
| 1 | 0.488124 | 15.5119 | 0.461916 | 14.703 |
| 4 | 0.494676 | 15.9141 | 0.486486 | 15.5972 |
| 6 | 0.501229 | 15.8922 | 0.476658 | 15.7315 |
| 8 | 0.473382 | 15.9247 | 0.474201 | 15.802 |

Table 16: SIRIUS on Sports.

| Experiment Settings treewidth | Llama-3-8B-Instruct-FSparse | | Llama-3-8B-Instruct-CSparse | |
|---|---|---|---|---|
| | Performance | AAL (out of 16) | Performance | AAL (out of 16) |
| Original Performance | 0.943299 | N/A | 0.943299 | N/A |
| Sparse Performance | 0.864948 | N/A | 0.879381 | N/A |
| 1 | 0.937113 | 12.3652 | 0.946392 | 9.95237 |
| 4 | 0.941237 | 14.5248 | 0.943299 | 11.5858 |
| 6 | 0.942268 | 14.8651 | 0.943299 | 14.0954 |
| 8 | 0.939175 | 14.9832 | 0.941237 | 14.7718 |
| | Llama-3-8B-FSparse | | Llama-3-8B-CSparse | |
| | Performance | AAL (out of 16) | Performance | AAL (out of 16) |
| Original Performance | 0.898969 | N/A | 0.898969 | N/A |
| Sparse Performance | 0.748454 | N/A | 0.720619 | N/A |
| 1 | 0.86653 | 15.5259 | 0.845361 | 13.5897 |
| 4 | 0.849485 | 15.5917 | 0.847423 | 15.2325 |
| 6 | 0.863918 | 15.5256 | 0.843299 | 15.4376 |
| 8 | 0.869072 | 15.6014 | 0.841237 | 15.5023 |
| | Llama-2-13B-Chat-FSparse | | Llama-2-13B-Chat-CSparse | |
| | Performance | AAL (out of 16) | Performance | AAL (out of 12) |
| Original Performance | 0.742268 | N/A | 0.742268 | N/A |
| Sparse Performance | 0.690722 | N/A | 0.584536 | N/A |
| 1 | 0.710309 | 13.9767 | 0.717659 | 7.72686 |
| 4 | 0.735052 | 14.9247 | 0.728953 | 10.7298 |
| | Llama-2-13B-FSparse | | Llama-2-13B-CSparse | |
| | Performance | AAL (out of 16) | Performance | AAL (out of 16) |
| Original Performance | 0.709278 | N/A | 0.709278 | N/A |
| Sparse Performance | 0.635052 | N/A | 0.558763 | N/A |
| 1 | 0.669072 | 15.4924 | 0.639175 | 14.3603 |
| 4 | 0.657732 | 15.9845 | 0.658763 | 15.48 |
| | Llama-2-7B-Chat-FSparse | | Llama-2-7B-Chat-CSparse | |
| | Performance | AAL (out of 16) | Performance | AAL (out of 16) |
| Original Performance | 0.731959 | N/A | 0.731959 | N/A |
| Sparse Performance | 0.652677 | N/A | 0.596907 | N/A |
| 1 | 0.704124 | 14.3861 | 0.712371 | 11.1517 |
| 4 | 0.712371 | 15.5904 | 0.71134 | 13.7394 |
| 6 | 0.709278 | 15.7475 | 0.71134 | 13.9857 |
| 8 | 0.698969 | 15.9927 | 0.715464 | 14.3817 |
| | Llama-2-7B-FSparse | | Llama-2-7B-CSparse | |
| | Performance | AAL (out of 16) | Performance | AAL (out of 16) |
| Original Performance | 0.545361 | N/A | 0.545361 | N/A |
| Sparse Performance | 0.536082 | N/A | 0.528866 | N/A |
| 1 | 0.524742 | 15.6754 | 0.536082 | 14.1031 |
| 4 | 0.547423 | 15.937 | 0.538144 | 15.6263 |
| 6 | 0.545361 | 15.9807 | 0.540206 | 15.7243 |
| 8 | 0.545361 | 15.9927 | 0.549485 | 15.811 |

Table 17: SIRIUS on Date.

| Experiment Settings treewidth | Llama-3-8B-Instruct-FSparse | | Llama-3-8B-Instruct-CSparse | |
|---|---|---|---|---|
| | Performance | AAL (out of 16) | Performance | AAL (out of 16) |
| Original Performance | 0.657224 | N/A | 0.657224 | N/A |
| Sparse Performance | 0.518414 | N/A | 0.532578 | N/A |
| 1 | 0.688385 | 14.2885 | 0.671388 | 14.6771 |
| 4 | 0.671388 | 15.357 | 0.685552 | 15.6324 |
| 6 | 0.679887 | 15.2435 | 0.688385 | 15.1663 |
| 8 | 0.674221 | 15.2654 | 0.694051 | 15.4293 |
| | Llama-3-8B-FSparse | | Llama-3-8B-CSparse | |
| | Performance | AAL (out of 16) | Performance | AAL (out of 16) |
| Original Performance | 0.583569 | N/A | 0.583569 | N/A |
| Sparse Performance | 0.399433 | N/A | 0.424929 | N/A |
| 1 | 0.535014 | 15.4236 | 0.535411 | 14.4364 |
| 4 | 0.543909 | 15.4782 | 0.546742 | 15.606 |
| 6 | 0.546742 | 15.6365 | 0.526912 | 15.7718 |
| 8 | 0.549575 | 15.7159 | 0.541076 | 15.7997 |
| | Llama-2-13B-Chat-FSparse | | Llama-2-13B-Chat-CSparse | |
| | Performance | AAL (out of 16) | Performance | AAL (out of 16) |
| Original Performance | 0.524079 | N/A | 0.524079 | N/A |
| Sparse Performance | 0.498584 | N/A | 0.419263 | N/A |
| 1 | 0.490085 | 13.9589 | 0.461756 | 14.1419 |
| 4 | 0.524079 | 15.432 | 0.478992 | 15.8545 |
| | Llama-2-13B-FSparse | | Llama-2-13B-CSparse | |
| | Performance | AAL (out of 16) | Performance | AAL (out of 16) |
| Original Performance | 0.501416 | N/A | 0.501416 | N/A |
| Sparse Performance | 0.464589 | N/A | 0.390935 | N/A |
| 1 | 0.447592 | 15.5992 | 0.461756 | 15.3896 |
| 4 | 0.492918 | 15.9129 | 0.484419 | 15.8357 |
| | Llama-2-7B-Chat-FSparse | | Llama-2-7B-Chat-CSparse | |
| | Performance | AAL (out of 16) | Performance | AAL (out of 16) |
| Original Performance | 0.320113 | N/A | 0.320113 | N/A |
| Sparse Performance | 0.339943 | N/A | 0.3002823 | N/A |
| 1 | 0.31728 | 14.4663 | 0.305949 | 5.75938 |
| 4 | 0.345609 | 15.6588 | 0.325779 | 14.7519 |
| 6 | 0.342776 | 15.742 | 0.314448 | 14.5768 |
| 8 | 0.348442 | 15.7692 | 0.308782 | 14.3627 |
| | Llama-2-7B-FSparse | | Llama-2-7B-CSparse | |
| | Performance | AAL (out of 16) | Performance | AAL (out of 16) |
| Original Performance | 0.33711 | N/A | 0.33711 | N/A |
| Sparse Performance | 0.314448 | N/A | 0.235127 | N/A |
| 1 | 0.342776 | 15.5144 | 0.269122 | 15.3598 |
| 4 | 0.342776 | 15.9141 | 0.266289 | 15.8553 |
| 6 | 0.328612 | 15.943 | 0.274788 | 15.9266 |
| 8 | 0.322946 | 15.9671 | 0.271955 | 15.956 |

Table 18: SIRIUS on StrategyQA.

| Experiment Settings treewidth | Llama-3-8B-Instruct-FSparse | | Llama-3-8B-Instruct-CSparse | |
|---|---|---|---|---|
| | Performance | AAL (out of 16) | Performance | AAL (out of 10) |
| Original Performance | 0.770241 | N/A | 0.770241 | N/A |
| Sparse Performance | 0.713348 | N/A | 0.562363 | N/A |
| 1 | 0.741794 | 8.98893 | 0.737418 | 6.60992 |
| 4 | 0.741794 | 9.48412 | 0.746171 | 7.92521 |
| 6 | 0.743982 | 9.53292 | 0.728665 | 8.22667 |
| 8 | 0.743982 | 9.55946 | 0.708972 | 8.97268 |
| | Llama-3-8B-FSparse | | Llama-3-8B-CSparse | |
| | Performance | AAL (out of 16) | Performance | AAL (out of 16) |
| Original Performance | 0.649891 | N/A | 0.649891 | N/A |
| Sparse Performance | 0.599562 | N/A | 0.439825 | N/A |
| 1 | 0.623632 | 9.46018 | 0.531729 | 8.68633 |
| 4 | 0.623632 | 9.74383 | 0.560175 | 9.44497 |
| 6 | 0.632385 | 9.83975 | 0.560175 | 9.58122 |
| 8 | 0.680525 | 9.80493 | 0.555799 | 9.67198 |
| | Llama-2-13B-Chat-FSparse | | Llama-2-13B-Chat-CSparse | |
| | Performance | AAL (out of 16) | Performance | AAL (out of 16) |
| Original Performance | 0.695842 | N/A | 0.695842 | N/A |
| Sparse Performance | 0.706783 | N/A | 0.634573 | N/A |
| 1 | 0.71116 | 9.48266 | 0.682713 | 6.74106 |
| 4 | 0.667396 | 9.83767 | 0.715536 | 8.09959 |
| 6 | 0.671772 | 9.88989 | 0.693654 | 8.82263 |
| | Llama-2-13B-FSparse | | Llama-2-13B-CSparse | |
| | Performance | AAL (out of 16) | Performance | AAL (out of 16) |
| Original Performance | 0.63895 | N/A | 0.63895 | N/A |
| Sparse Performance | 0.693654 | N/A | 0.533917 | N/A |
| 1 | 0.695842 | 9.8979 | 0.595186 | 8.77388 |
| 4 | 0.682713 | 9.96438 | 0.643326 | 9.47368 |
| 6 | 0.689278 | 9.9789 | 0.63895 | 9.56319 |
| | Llama-2-7B-Chat-FSparse | | Llama-2-7B-Chat-CSparse | |
| | Performance | AAL (out of 16) | Performance | AAL (out of 16) |
| Original Performance | 0.654267 | N/A | 0.654267 | N/A |
| Sparse Performance | 0.678337 | N/A | 0.612691 | N/A |
| 1 | 0.684902 | 9.64754 | 0.669584 | 6.55818 |
| 4 | 0.691466 | 9.79539 | 0.671772 | 7.88988 |
| 6 | 0.68709 | 9.86474 | 0.643326 | 8.28982 |
| 8 | 0.689278 | 9.86488 | 0.66302 | 8.43513 |
| | Llama-2-7B-FSparse | | Llama-2-7B-CSparse | |
| | Performance | AAL (out of 16) | Performance | AAL (out of 16) |
| Original Performance | 0.599562 | N/A | 0.599562 | N/A |
| Sparse Performance | 0.592998 | N/A | 0.538293 | N/A |
| 1 | 0.612691 | 9.73256 | 0.568928 | 8.38473 |
| 4 | 0.599562 | 9.93662 | 0.560175 | 9.36272 |
| 6 | 0.617068 | 9.95582 | 0.536105 | 9.35857 |
| 8 | 0.610503 | 9.96658 | 0.544858 | 9.4642 |

Table 19: SIRIUS on HumanEval.

| Experiment Settings treewidth | Llama-3-8B-Instruct-FSparse | | Llama-3-8B-Instruct-CSparse | |
|---|---|---|---|---|
| | Performance | AAL (out of 16) | Performance | AAL (out of 16) |
| Original Performance | 0.560975609756098 | N/A | 0.560975609756098 | N/A |
| Sparse Performance | 0.457317073170732 | N/A | 0.207317073170732 | N/A |
| 1 | 0.585365853658537 | 14.7624 | 0.554878048780488 | 12.1326 |
| 4 | 0.579268292682927 | 15.2299 | 0.530487804878049 | 14.0546 |
| 6 | 0.615853658536585 | 15.4209 | 0.518292682926829 | 14.4431 |
| 8 | 0.585365853658537 | 15.5009 | 0.524390243902439 | 14.6725 |
| | Llama-3-8B-FSparse | | Llama-3-8B-CSparse | |
| | Performance | AAL (out of 16) | Performance | AAL (out of 16) |
| Original Performance | 0.26219512195122 | N/A | 0.26219512195122 | N/A |
| Sparse Performance | 0.189024390243902 | N/A | 0.0670731707317073 | N/A |
| 1 | 0.231707317073171 | 15.1878 | 0.109756097560976 | 12.1402 |
| 4 | 0.274390243902439 | 15.2827 | 0.219512195121951 | 13.7718 |
| 6 | 0.26219512195122 | 14.5355 | 0.207317073170732 | 14.7776 |
| 8 | 0.29268 | 15.6305 | 0.24390243902439 | 15.1074 |
| | Llama-2-13B-Chat-FSparse | | Llama-2-13B-Chat-CSparse | |
| | Performance | AAL (out of 16) | Performance | AAL (out of 12) |
| Original Performance | 0.189024390243902 | N/A | 0.189024390243902 | N/A |
| Sparse Performance | 0.146341463414634 | N/A | 0.121951219512195 | N/A |
| 1 | 0.170731707317073 | 14.3976 | 0.189024390243902 | 9.6447 |
| 4 | 0.182926829268293 | 15.1956 | 0.176829268292683 | 10.7946 |
| 6 | 0.182926829268293 | 15.3494 | 0.170731707317073 | 11.0149 |
| 8 | 0.176829268292683 | 15.4067 | 0.170731707317073 | 11.1252 |
| | Llama-2-13B-FSparse | | Llama-2-13B-CSparse | |
| | Performance | AAL (out of 16) | Performance | AAL (out of 16) |
| Original Performance | 0.176829268292683 | N/A | 0.176829268292683 | N/A |
| Sparse Performance | 0.158536585365854 | N/A | 0.0975609756097561 | N/A |
| 1 | 0.146341463414634 | 15.2129 | N/A | N/A |
| 4 | 0.158536585365854 | 15.9093 | 0.146341463414634 | 14.1813 |
| 6 | 0.170731707317073 | 15.9211 | 0.134146341463415 | 14.5866 |
| 8 | 0.176829268292683 | 15.9015 | 0.134146341463415 | 14.7508 |
| | Llama-2-7B-Chat-FSparse | | Llama-2-7B-Chat-CSparse | |
| | Performance | AAL (out of 16) | Performance | AAL (out of 12) |
| Original Performance | 0.140243902439024 | N/A | 0.140243902439024 | N/A |
| Sparse Performance | 0.134146341463415 | N/A | 0.0670731707317073 | N/A |
| 1 | 0.134146341463415 | 14.055 | 0.140243902439024 | 8.83176 |
| 4 | 0.146341463414634 | 14.8504 | 0.146341463414634 | 10.1263 |
| 6 | 0.152439024390244 | 15.1924 | 0.152439024390244 | 10.5576 |
| 8 | 0.164634146341463 | 15.2742 | 0.158536585365854 | 10.822 |
| | Llama-2-7B-FSparse | | Llama-2-7B-CSparse | |
| | Performance | AAL (out of 16) | Performance | AAL (out of 16) |
| Original Performance | 0.115853658536585 | N/A | 0.115853658536585 | N/A |
| Sparse Performance | 0.115853658536585 | N/A | 0.0792682926829268 | N/A |
| 1 | 0.115853658536585 | 15.5268 | 0.121951219512195 | 12.6604 |
| 4 | 0.128048780487805 | 15.8167 | 0.121951219512195 | 14.4053 |
| 6 | 0.164634146341463 | 15.8615 | 0.121951219512195 | 14.8296 |
| 8 | 0.109756097560976 | 15.9189 | 0.128048780487805 | 14.8443 |

Table 20: SIRIUS on MBPP+.

| Experiment Settings treewidth | Llama-3-8B-Instruct-FSparse | | Llama-3-8B-Instruct-CSparse | |
|---|---|---|---|---|
| | Performance | AAL (out of 16) | Performance | AAL (out of 16) |
| Original Performance | 0.584656084656085 | N/A | 0.584656084656085 | N/A |
| Sparse Performance | 0.531746031746032 | N/A | 0.248677248677249 | N/A |
| 1 | 0.537037037037037 | 14.7267 | 0.563492063492064 | 11.5415 |
| 4 | 0.563492063492064 | 15.2699 | 0.566137566137566 | 13.5896 |
| 6 | 0.552910052910053 | 15.3782 | 0.571428571428571 | 14.0547 |
| 8 | 0.552910052910053 | 15.4689 | 0.566137566137566 | 14.7648 |
| | Llama-3-8B-FSparse | | Llama-3-8B-CSparse | |
| | Performance | AAL (out of 16) | Performance | AAL (out of 16) |
| Original Performance | 0.518518518518519 | N/A | 0.518518518518519 | N/A |
| Sparse Performance | 0.433862433862434 | N/A | 0.161375661375661 | N/A |
| 1 | 0.4894 | 14.8849 | 0.415343915343915 | 12.7016 |
| 4 | 0.484126984126984 | 15.4346 | 0.407407407407407 | 14.0936 |
| 6 | 0.473544973544974 | 15.3581 | 0.433862433862434 | 14.5662 |
| 8 | 0.468253968253968 | 15.6088 | 0.41005291005291 | 14.4752 |
| | Llama-2-13B-Chat-FSparse | | Llama-2-13B-Chat-CSparse | |
| | Performance | AAL (out of 16) | Performance | AAL (out of 16) |
| Original Performance | 0.23015873015873 | N/A | 0.27 | N/A |
| Sparse Performance | 0.19047619047619 | N/A | 0.1 | N/A |
| 1 | 0.201058201058201 | 13.8235 | 0.26 | 9.32827 |
| 4 | 0.232804232804233 | 14.8394 | 0.26 | 10.7346 |
| 6 | 0.224867724867725 | 15.0801 | 0.26 | 10.8897 |
| 8 | 0.227513227513228 | 15.2373 | 0.25 | 11.0214 |
| | Llama-2-13B-FSparse | | Llama-2-13B-CSparse | |
| | Performance | AAL (out of 16) | Performance | AAL (out of 16) |
| Original Performance | 0.246031746031746 | N/A | 0.21 | N/A |
| Sparse Performance | 0.232804232804233 | N/A | 0.13 | N/A |
| 1 | 0.214285714285714 | 14.8374 | 0.22 | 14.5174 |
| 4 | 0.259259259259259 | 15.8547 | 0.24 | 15.6461 |
| 6 | 0.235449735449735 | 15.9197 | 0.23 | 15.7174 |
| 8 | 0.246031746031746 | 15.9094 | 0.22 | 15.7179 |
| | Llama-2-7B-Chat-FSparse | | Llama-2-7B-Chat-CSparse | |
| | Performance | AAL (out of 16) | Performance | AAL (out of 12) |
| Original Performance | 0.261904761904762 | N/A | 0.261904761904762 | N/A |
| Sparse Performance | 0.224867724867725 | N/A | 0.100529100529101 | N/A |
| 1 | 0.238095238095238 | 14.1571 | 0.214285714285714 | 8.61325 |
| 4 | 0.26984126984127 | 14.9264 | 0.23015873015873 | 10.2517 |
| 6 | 0.238095238095238 | 15.2194 | 0.227513227513228 | 10.5845 |
| 8 | 0.272486772486773 | 15.3086 | 0.235449735449735 | 10.7621 |
| 10 | N/A | N/A | 0.232804232804233 | 10.8962 |
| | Llama-2-7B-FSparse | | Llama-2-7B-CSparse | |
| | Performance | AAL (out of 16) | Performance | AAL (out of 12) |
| Original Performance | 0.253968253968254 | N/A | 0.253968253968254 | N/A |
| Sparse Performance | 0.201058201058201 | N/A | 0.0793650793650794 | N/A |
| 1 | 0.216931216931217 | 14.6103 | 0.171957671957672 | 10.6643 |
| 4 | 0.238095238095238 | 15.5672 | 0.185185185185185 | 11.5561 |
| 6 | 0.224867724867725 | 15.6273 | 0.195767195767196 | 11.6547 |
| 8 | 0.240740740740741 | 15.5569 | 0.203703703703704 | 11.6753 |

