# OpenReview forum: "SIRIUS : Contexual Sparisty with Correction for Efficient LLMs"
_NeurIPS.cc/2024/Conference — NeurIPS 2024 poster_

### Official Review · Reviewer_BB9w · 2024-06-26

**Soundness:** 2
**Presentation:** 2
**Contribution:** 2
**Rating:** 6
**Confidence:** 4

**Summary:**

The paper introduces a sparse LLM correction mechanism (SIRIUS) designed to improve the inference efficiency of large language models through contextual sparsity. Contextual sparsity reduces computational cost by pruning parameters that are less relevant based on the input context. However, this approach degrades performance in tasks requiring high-level reasoning and deduction. SIRIUS addresses this by selectively correcting key tokens, using the full model to rewrite the sparse model's Key-Value cache. Experiments show improvements in strict-match EM scores, particularly on mathematical reasoning tasks.

**Strengths:**

1. The correction mechanism that selectively rewrites the KV cache is novel.
2. The paper presents an interesting finding where sparse models fail on tasks that require reasoning and deduction.
3. The paper demonstrates improvements in performance metrics on benchmarks like GSM8K.

**Weaknesses:**

- The KV cache rewriting mechanism requires an available full LLM to be of the same architecture as the sparse model.
- The criteria for when to trigger KV cache rewriting are not clearly detailed, making it difficult to assess the practicality and reliability of the approach.
- It is unclear whether using the full model’s KV cache to rewrite the sparse model guarantees that the rewritten KV cache can be accurately interpreted and utilised by the sparse model, due to the differences in the sparse and full model.
- The hypothesis that the gap between the full and sparse models can be bridged by correcting a few key tokens (L44-45) is not sufficiently validated beyond the provided examples. It remains unclear whether this hypothesis holds across a broader range of tasks and datasets.
- The writing needs to be improved for clarity, for example,
     - there are a few missing citations (e.g., L114. L284)
     - the tables can be clarified with the best performing variant per task in boldface, and a caption which describes the metrics in the table.
     - Algorithm 1's functions can be better described, for example, the descriptions in "normal forward function FORWARD, forward function with direct cache rewrite KVFORWARD, likelihood judge function of LIKELIHOOD" are unclear.

**Questions:**

- How are the likelihood thresholds for detecting and correcting errors empirically tuned? Can the authors provide more details on this process and its impact on performance?
- How does SIRIUS handle long-term dependencies and corrections over extended sequences of text? Are there scenarios where the KV cache rewriting might not be sufficient to correct extended errors?

**Limitations:**

not observed negative societal impact to my understanding.

---

> ### Author Rebuttal · Authors · 2024-08-07
>
> Thank you so much for your time and attention on the paper. We are very grateful for you to be interested in the KV Cache correction. We will try our best to answer your questions.
>
> 1. Full and Sparse of the same architecture
>
> Contextual Sparsity is the focus of Sirius. Contextually Sparse (CS) models are dynamically subsampled from the normal LLM. The subsampling process won't change the compatibility of the original model's cache with the newer subsampled sparse model. Therefore, the original cache is naturally usable by CS models. However, we are interested to see whether Sirius can be applied to model pairs outside the CS realm.
>
> 2. When to rewrite KV Cache
>
> We appreciate your suggestions, and we rewrote the method part of the paper to take into consideration KV rewriting confusion. The rewriting happens when the full model is correcting the sparse generated tokens. The full model is called once every kernel size, usually 16. Throughout the inference, the KV cache is shared between the sparse and the full model. The KV cache is mostly populated by the sparse model, which is called for every token. During correction, the full model takes in the last kernel size of tokens and generates its KVs for the past 16 tokens in parallel, these KVs are directly written to their corresponding positions in the shared KV Cache.
>
> 3. Will the full model's KVs make semantic sense to the sparse model
>
> We appreciate the brilliant question. Empirically, we found that the KV cache of the large model seems to always help the small model generation quality. Below shows one of the example on GSM8K (20% subsampled)
> | GSM8K 20%                                 |          | GSM8K 20%                                           |          |
> |-------------------------------------------|----------|-----------------------------------------------------|----------|
> | Llama3-8B-Instruct                        | 0.7538/0.7538 | Llama3-8B-Instruct                                  | 0.7538/0.7538 |
> | Llama3-8B-Instruct + Griffin              | 0.3674/0.3674 | Llama3-8B-Instruct + CATS                           | 0.5644/0.5644 |
> | Llama3-8B-Instruct + Griffin + KV Cache Correct | 0.4735/0.4735 | Llama3-8B-Instruct + CATS + KV Cache Correct        | 0.6629/0.6629 |
> You can see that doing KV Cache correction alone brings substantial improvement to both settings. Empirically, the sparse model is clearly able to extract more insightful semantics from the full model's KVs.
>
> 4. Whether correcting minor portion leads to performance recovery is applicable to more datasets
>
> We appreciate the comments. Although we only show for GSM8K in the text. However, in the newer version of the paper. We evaluated Sirius across diverse datasets, ranging from arithmetic reasoning, common sense reasoning, and coding. Sirius managed to achieve competitive efficiency metrics on most of them. These results show that for general text generation that requires challenging step-by-step correlation, the minor portion is the only part that needs to be corrected to recover the original performance.
>
> 5. Writing is unclear
>
> We deeply appreciate the suggestions on improving the writing quality of the paper. All of the suggested parts are revised in the newer version of the paper.
>
> 6. How to determine thresholds.
>
> Threshold determination is a crucial process of balancing the performance and efficiency of Sirius. Here we provide a much more in-depth ablation on how increasing the threshold exhibits the trend that improves the performance but hurts the efficiency. The efficiency of Sirius is measured by the average advance length out of a kernel size (usually 16 tokens), the higher the better.
> | Threshold                     | Performance Score | Efficiency Metrics |
> |-------------------------------|-------------------|--------------------|
> | Original Full Model           | 0.7803/0.7828     |                    |
> | No correction (threshold 0)   | 0.5884/0.5960     |                    |
> | 0.05                          | 0.7247/0.7247     | 15.2098            |
> | 0.1                           | 0.7399/0.7424     | 14.6451            |
> | 0.2                           | 0.7247/0.7273     | 13.2329            |
> | 0.3                           | 0.7399/0.7449     | 11.6134            |
> | 0.4                           | 0.7551/0.7601     | 10.0037            |
> | 0.5                           | 0.7677/0.7702     | 8.56022            |
> | 0.6                           | 0.7758/0.7778     | 7.44126            |
> | 0.7                           | 0.7702/0.7753     | 6.26547            |
> | 0.8                           | 0.7753/0.7803     | 5.25639            |
> | 0.9                           | 0.7652/0.7677     | 4.20542            |
> | 0.95                          | 0.7626/0.7652     | 3.56315            |
> | 1.0                           | 0.7601/0.7626     | 1.2685             |
> In practice, threshold of 0.1 generally works well across various models and datasets.
>
> 7. Long range dependencies
>
> Sirius allows the large model to corrects the small model on a basis of kernel size. Besides KV rewriting, roll back mechanism is especially important for generating longer text. The rollback works as follows, when full model parallel verifies, If there is any token within the range of the kernel size that the full model regards as unlikely (rejects by likelihood threshold), then the token will be removed. Therefore, regardless of the generation length, all the verified tokens will be regarded by the full model as "enough likely" given the context.

---

> > ### Comment · Reviewer_BB9w · 2024-08-13
> >
> > The rebuttal addresses some of my concerns. I have increase my score to 6.

---

> > > ### Author Response · Authors · 2024-08-14
> > > **Thank you for your comment**
> > >
> > > We would like to express our deep appreciation for your time reading our work. Your insightful comments on KV cache rewrite mechanism really help us reflect on our previous description of Sirius, and it motivates us to refine it for more readers to understand more easily. Thank you for your comments.

---

### Official Review · Reviewer_ArEw · 2024-07-11

**Soundness:** 2
**Presentation:** 2
**Contribution:** 2
**Rating:** 5
**Confidence:** 3

**Summary:**

The paper introduces SIRIUS, a novel method designed to enhance the efficiency and accuracy of sparse Large Language Models in reasoning and deduction tasks. SIRIUS employs contextual sparsity to reduce computational costs and incorporates an efficient correction mechanism that recovers sparse model performance by correcting only a minimal number of key tokens. Experimental results demonstrate that SIRIUS significantly improves the performance of sparse models on complex reasoning tasks while maintaining efficiency.

**Strengths:**

1.	The paper introduces SIRIUS, a novel and efficient correction mechanism specifically tailored for sparse models
2.	It provides a echnically rigorous approach to improving the inference efficiency of LLMs, supported by experimental validation.

**Weaknesses:**

1.	There are too few experiments,  so no experiments were conducted on other LLMs， and no experiments were conducted on more contextual sparsity methods
2.	The Mc annotation in pseudocode is unclear

**Questions:**

If the correction requires the use of a full model, won't the speed of correction slow down compared to directly using the sparse model? Is the memory usage the same as using a full model directly? If the memory usage is the same as using a full model directly, there is no advantage in memory savings

**Limitations:**

The author improved the performance of the sparse model in reasoning and understanding tasks by adding a full model correction, which requires the use of the entire full model. Although the average parameters used per token increase is minimal, the memory usage is the same as using the full model directly

---

> ### Author Rebuttal · Authors · 2024-08-07
>
> Thank you for the time taken to read the paper and your insightful comments. We are thankful that you suggested we add more experiment data to evaluate Sirius. We will try our best to address your concern. If you feel that your concern has been taken into account, please consider raising your score.
>
> 1. Too few empirical experiments
>
> We provide additional empirical results and analysis of Sirius on various models and different downstream datasets.
>
> As discussed in the paper, the weakness of the contextual sparse models seems to occur upon difficult reasoning tasks. Besides, we collect experiment results on different reasoning tasks. In Arithmetic reasoning, besides GSM-8K, we also show a difficult AQuA-RAT COT from Google. In the Common Sense reasoning, we follow the COT paper to evaluate CSQA, StrategyQA, Sports, and Dates. Also, we found that contextual sparse models cannot do well in code generation. We then run Sirius on HumanEval.
>
> We collect additional experiment results on six models (Llama-3-8B, Llama-2-7B, Llama-2-13B with their instruction fine-tuning counterparts). Due to the word limit, we only show AQuA, CSQA, and HumanEval on some models. Full results and Average advance length will be in the new version of the paper.
> L means Llama.
> O means Original
> S means Sparse
> SS means Sparse + Sirius
>
> | AQuA RAT                          | L-3-8B-I-FSparse | L-3-8B-I-CSparse | L-2-7B-Chat-FSparse | L-2-7B-Chat-CSparse |
> |-----------------------------------|-----------------------------|-----------------------------|-------------------------|-------------------------|
> | O               | 0.51                        | 0.51                        | 0.25                    | 0.25                    |
> | S                | 0.42                        | 0.27                        | 0.28                    | 0.22                    |
> | SS                            | 0.42                        | 0.46                        | 0.24                    | 0.25                    |
> | |L-2-13B-Chat-FSparse          | L-2-13B-Chat-CSparse    |         |      |
> | O             | 0.23                        | 0.23                        |                     |                     |
> | S               | 0.25                        | 0.20                        |                     |                     |
> | SS                            | 0.27                        | 0.26                        |                     |                    |
>
> | CSQA                                | L-3-8B-I-FSparse | L-3-8B-I-CSparse | L-2-7B-Chat-FSparse | L-2-7B-Chat-CSparse |
> |-------------------------------------|-----------------------------|-----------------------------|-------------------------|-------------------------|
> | O               | 0.70                        | 0.70                        | 0.62                    | 0.62                    |
> | S                 | 0.61                        | 0.64                        | 0.61                    | 0.52                    |
> | SS                              | 0.69                        | 0.72                        | 0.63                    | 0.60                    |
> |                                     | L-2-13B-Chat-FSparse    | L-2-13B-Chat-CSparse    |      |      |
> | O                | 0.68                        | 0.68                        |                     |                     |
> | S                 | 0.53                        | 0.55                        |                     |                     |
> | SS                              | 0.65                        | 0.67                        |                     |                     |
>
> | HumanEval                          | L-3-8B-I-FSparse | L-3-8B-I-CSparse | L-2-7B-Chat-FSparse | L-2-7B-Chat-CSparse |
> |------------------------------------|-----------------------------|-----------------------------|-------------------------|-------------------------|
> | O             | 0.56                        | 0.56                        | 0.14                    | 0.14                    |
> | S                 | 0.45                        | 0.20                        | 0.13                    | 0.07                    |
> | SS                             | 0.58                        | 0.55                        | 0.13                    | 0.15                    |
> |                                    | L-2-13B-Chat-FSparse    | L-2-13B-Chat-CSparse    |      |      |
> | O               | 0.18                        | 0.18                        |                     |                     |
> | S                | 0.14                        | 0.12                        |                     |                     |
> | SS                             | 0.17                        | 0.17                        |                     |                     |
>
> Sirius can also work effectively on the Llama-3-70B-Instruct, model with large parameters.
> | L-3-70B-I                              |     GSM8KCOT           |
> |---------------------------------------------------|----------------|
> | L-3-70B-I                               | 0.90  |
> | L-3-70B-I CSparse         | 0.74  |
> | L-3-70B-I CSparse         | 0.87   (15.41)
> The bracket is the average advance length of 16 kernel size.
>
>
> 2. Other questions.
>
> a. We appreciate the suggestions on Pseudo code, we have rewritten it in the newer version.
>
> b. Correction step by full model is slower than sparse, but the cost is amortized by the kernel size of 16, which is less significant.
>
> c. Contextual sparsity subsample sparse model from the full model dynamically according to input, so the full model weight will always be in GPU memory. Adding correction won't increase the memory usage of the sparse model.
>
> d. Please note that the latency is mainly affected by the memory loading not the shear memory size on VRAM. Full model is loaded only once per kernel size. Please refer to the hardware speedup on the new version.

---

> ### Author Response · Authors · 2024-08-13
> **Kindly Asking to Reconsider Your Score**
>
> Thank you again for your feedback.
>
> We have added the necessary material to the author's rebuttal to address the points you raised. Given that we only have four reviewers, each reviewer's score significantly impacts the overall assessment of our work. We believe that the current overall assessment does not adequately reflect the contribution of our work. Therefore, we kindly request you reconsider your score. Thank you again for your time and effort in reviewing our paper.

---

> > ### Comment · Reviewer_ArEw · 2024-08-13
> > **Response by Reviewer ArEw**
> >
> > Thank you for your detailed explanation and rich experimental supplements. It alleviated my concerns to a certain extent. I raise my score to 5.

---

> > > ### Author Response · Authors · 2024-08-14
> > > **Thank you for your comments**
> > >
> > > We would like to express our deep appreciation for your time reading our work. The comment on lacks of empirical results help us make our study on Contextual Sparsity correction more concrete.

---

### Official Review · Reviewer_rbBk · 2024-07-11

**Soundness:** 1
**Presentation:** 1
**Contribution:** 2
**Rating:** 6
**Confidence:** 4

**Summary:**

This paper aims to improve contextual sparsity (CS) approaches: LLM sparsification / parameter pruning methods where the sparsification strategy is conditioned on the input sequence / prompt itself. The paper begins by reproducing contextual sparsity baselines, but applied to modern LLMs: LLama 2 and 3 (7B & 8B respectively), and verifying that their reproduction successfully replicates previous CS techniques on summarization and question answering tasks. The authors then demonstrate that the technique is much less effective on more challenging reasoning tasks such as GSM8K and MMLU-CoT. Through investigation the authors observe that often, by correcting a small amount of tokens during the generation (~10%), the sparse models can significantly improve their performance to be on-par with the full dense model. This motivates their main proposed technique: SIRIUS, which employs a speculative decoding like technique to use the full dense model to score every k tokens (k=16-24 in practice) during the decoding process in order to rollback and use the dense model to correct a token when necessary. By doing so, the method is able to mitigate the losses on reasoning tasks while maintaining most of the efficiency of the sparsification method.

**Strengths:**

1. The authors explore the viability of CS techniques on much more difficult reasoning tasks than previously in the literature, providing an important datapoint on whether CS approaches (at least ones studied in this paper) work in general.

1. The authors run thorough quality experiments on modern LLMs, and demonstrate that their technique effectively rectifies the issues they first identify (significantly improving reasoning performance.)

1. Using the full model to do something like self-speculative decoding with the drafting model being a sparse version of the same model is novel and promising. It aligns with contemporaneous works that explore this idea [1].

1. The method is efficient w.r.t the number of parameters used on average during decoding.

1. The paper is overall well structured and the investigation is easy to follow.

1. The authors plan to release the code for their experiments.

[1] LayerSkip: Enabling Early Exit Inference and Self-Speculative Decoding

**Weaknesses:**

1. The main spirit of this paper is whether we can use some small amount of additional compute budget over the sparse model alone (e.g. via limited calls to the full model) to improve the decoding from the sparse model. However, the authors do not compare against any baseline approaches to do this same thing, such as speculative decoding. Even though the authors claim in Related Work that speculative decoding does not suit this problem since it’s not efficient enough, the authors do not provide any empirical evidence of such.

1. This paper is very much an efficiency paper, which tries to introduce some small amount of corrections from the full model to the dense model, while not doing too much as to nullify the gains from the sparse method. However, the authors only analyze Average Parameters Used Per Token (APU), a sparsity measure, instead of any other efficiency metric (importantly wall-time).
    - The authors make an argument for why this is in Section 4.1, using findings from previous papers. While this may be a reasonable intuitive argument, the method in this paper has significant differences like KV-cache correction that still makes this valuable to empirically verify.
    - Moreover, even if I intuitively believe from the author’s argument that the wall time latency of this approach would be somewhere between the sparse and full model’s wall time, it is unclear where on this spectrum it lies, how it compares to other techniques like speculative decoding, or how the tradeoff plays with varying sparsity levels without such measurement and comparison.
    - To place this into context, the CATS paper, which is one of the methods this work experiments with (FSparse) directly states “Activation sparsity of a model is not sufficient to directly enable wall-clock time inference
Speedups”, and provides wall-clock time analysis.

1. The paper is poorly written, with various spelling errors (in the title), grammatical errors (for example: line 1, Section 4.2 title, and more), placeholders for citations (line 114), and difficult to read citation formatting. In many ways the presentation of this work is significantly not publication ready.

1. Without the code release, the description of the method is not very detailed, even though Algorithm 1 is provided, and would make it very hard for the reader to understand how this method is implemented from reading the paper alone, which hurts reproducibility.
    1. For example: the description of the correction algorithm is only described in one paragraph in Section 4.3 with insufficient detail on how the KV cache is updated.
    1. Algorithm 1 has undefined variables with respect to the KV cache such as “cachetracker”. The description in 4.3 states that the KV cache is directly written to by the full model to correct the past, however Algorithm 1 does not describe this process other than  “Update C_s based on j” is unclear what kind of update is happening to the cache.
    1. The important LIKELIHOOD function is undefined, which uses the full model’s likelihoods to judge whether the sparse model’s output is incorrect. There are no details on what threshold is used here.

1. Some additional claims are not sound:
    - Line 270: “KV Cache can also be looked as an implicit knowledge distillation method to pass dark information from the LLM to the sparse counterparts” What is dark information?

**Questions:**

What were the motivations behind the differences between Sirius (Algorithm 1 / Section 4.3) and speculative decoding? Why not directly use speculative decoding, with the sparse model being the drafting model? (which maybe has more guarantees)

Why is the method named Sirius?

**Limitations:**

The authors briefly describe some limitations but the work is generally lacking discussion here. Mainly the authors mention that the method does not work well under extreme sparsity (to my knowledge almost no sparsity method does in the general case – this is more of an open question than a limitation.) The authors could be more forthcoming here. It would be valuable to elaborate on whether the authors believe the two CS approaches are representative of the space of approaches, whether the experiments are adequate, whether they expect the results to change at different model scales, or for tasks beyond the ones studied here.

---

> ### Author Rebuttal · Authors · 2024-08-07
>
> Thank you for the time taken to read the paper and your insightful comments. We are thankful that you point out that the comparison against speculative decoding is missing and we lack of convincing real efficiency metrics.  We will try our best to address your concern. If you feel that your concern has been taken into account, please consider raising your score.
>
> 1. Differences to Speculative Decoding? Why Speculative Decoding is not efficient enough?
>
> Sirius contains a stronger and a weaker model, the strong verifies the weaker's output, from which the resemblance can be easily drawn between Sirius and Speculative Decoding (SD). However, the important distinction is the problem setting.
>
> Contextual sparsity users prioritize the tradeoff between the performance and the efficiency of the model. We show that the efficiency of the contextually sparse model corrected by SD will be largely limited. Take the Coarse-grained sparse model on Llama-3-8B as an example, the 50% sparse (APU 0.65) model will produce an acceptance rate of 0.89 on GSM8K. SD also runs the smaller model every iteration with verification of the larger model once a period. Naturally, to increase efficiency, we need to (1) enlarge the period and (2) increase the average number of tokens accepted in the period.
> \frac{1 - \alpha^{(\gamma + 1)}}{1 - \alpha}
> Now, given the acceptance rate of 0.89, following the formulation in Speculative decoding, we can calculate the expected number of accepted tokens for every gamma term in the Speculative Decoding literature, which is (period - 1).
>
> |   Gamma |   Advance Length |
> |--------:|-----------------:|
> |       4 |          4.01    |
> |       8 |          5.91    |
> |      12 |          7.09    |
> |      16 |          7.84    |
> |      20 |          8.30    |
> |      24 |          8.60    |
> |      28 |          8.78    |
> |      32 |          8.90    |
>
> We can notice the trend that the average advance length starts to plateau as the period becomes larger. Take the gamma of 16 as an example, the period is then 17. If we use the formula in Section 4.1 of our paper, we can immediately see that the APU is (16 * 0.65 + 1)/7.84 = 1.45, even larger than full model 1.
>
> |   Gamma |   Advance Length |
> |--------:|-----------------:|
> |       1 |           0.87   |
> |       2 |           0.86   |
> |       4 |           0.90   |
> |       8 |           1.05   |
> |      12 |          1.24    |
> |      16 |          1.45    |
> |      20 |          1.69    |
> |      24 |          1.93    |
>
> Because of the plateauing effect, for an acceptance rate of 0.89, the best gamma is 2 (period = 3). The best efficiency is 0.86, compared with 0.65 coarse-grained sparse APU. A similar picture can be applied to Fine-grained sparsity as well. The key reasons are (1) the contextual sparse model is too large to use a long period for SD, and (2) the acceptance criteria are too strict.
>
> In contrast, Sirius gives a more flexible choice for contextually sparse model users. For a threshold of 0.1, Sirius can correct Llama-3-8B coarse-grained sparsity from 20.85% to 43.9%, compared to the 49.66% full model. Sirius on average can accept 13.4 tokens out of a kernel size of 16 and over 9 tokens out of a kernel size of 10, translating to APU < 0.76, significantly lower than SD does.
>
> 2. Hardware speedup?
>
> We show that Sirius delivers the promised speedup.
> With the introduction of static cache from huggingface 4.38, kv cache is pre-allocated, making rewrite, expand, and rollback of very minimal overhead. Fine-grained sparsity speedup is via prior work [1]. However, CATS relies on a close-sourced custom CUDA kernel. We present the speedup of Coarse-grained sparsity with Sirius in two settings.
> First, on-chip, we run Llama-3-8B-Instruct inference.
>
> | Setting| GSM-8K-COT | A40 | Speedup | L40 | Speedup | A100 | Speedup |
> |-----|-----|-----|-----|-----|-----|-----|-----|
> | CSparsity | 0.3601 | 20.7 | 0.66 | 15.6 | 0.67 | 9.6 | 0.72 |
> | Sirius (10 kernel size) |0.7309| 24.1 | 0.78 | 18.2 | 0.78 | 11.4 | 0.85 |
> | Sirius (16 kernel size) | 0.7309 | 24.9 | 0.80 | 18.8 | 0.81 | 11.8 | 0.88 |
> | Full | 0.7612 | 30.9 | | 23.2 | | 13.3 | |
>
> Second, Offloading Llama-3-70B-Instruct. We use a single L40 48GB with a PCIe bus bandwidth of 25 GB/s.
>
> | | Sparse | Sparse + Sirius | Full |
> |---------------|----------------------|-------------------------------|--------------------|
> | Performance | 0.9014 | 0.7407 | 0.7407/0.7483 | |
> | Latency (s) | 3.57 | 3.68 | 5.72 | 25.3 GB/s |
> | Ratio to Full | 0.6241 | 0.6434 | | |
>
> [1] Liu, Z., Wang, J., Dao, T., Zhou, T., Yuan, B., Song, Z., ... & Chen, B. (2023, July). Deja vu: Contextual sparsity for efficient llms at inference time. In International Conference on Machine Learning (pp. 22137-22176). PMLR.
>
> 3. Criticism of writing quality
> We do appreciate the suggestions on writing. In light of your precious effort put into reading, we carefully rewrote and proofread the entire methodology section that takes all your suggestions into account.
>
> 4. KV Cache correction
> | Setting|GSM8K 20%                              |Setting| GSM8K 20%                                           |
> |-------------------------------------------|----------|-----------------------------------------------------|----------|
> | Llama3-8B-Instruct                        | 0.7538 | Llama3-8B-Instruct                                  | 0.7538 |
> | Llama3-8B-Instruct + CSparse             | 0.3674 | Llama3-8B-Instruct +     FSparse                       | 0.5644 |
> | Llama3-8B-Instruct + CSparse + KV Cache Correct | 0.4735 | Llama3-8B-Instruct + FSparse + KV Cache Correct        | 0.6629 |
> We show that KV Cache correction contributes to correction. However, we agree that "implicit distillation" is unbased, which has been deleted in the latest version.
>
> 5. Why Sirius?
> Sirius is an astronomical term referring to a two-body star system. One is the brightest ever known, while the other is dim. We draw inspiration from that system.

---

> > ### Comment · Reviewer_rbBk · 2024-08-12
> > **Updated rating**
> >
> > Thank you for providing your extensive clarifications and updated results. In particular, the latency evaluations on real hardware, additional ablations, and revised presentation changes my impression of this paper a lot. I have increased my score to 6.
> >
> > Nit: there is still a typo in the title "Sparisty", and other places like Figure 3 title "Efficientcy". Please continue to refine for the future versions.

---

> > > ### Author Response · Authors · 2024-08-14
> > > **Thank you for your careful reading and raising the score**
> > >
> > > We would like to express our deep appreciation for your careful reading and insightful comments. Your comments hit home on the important insight of Sirius. We really appreciate your time and effort spent on our work.

---

### Official Review · Reviewer_a8uY · 2024-07-13

**Soundness:** 2
**Presentation:** 3
**Contribution:** 2
**Rating:** 4
**Confidence:** 4

**Summary:**

This paper focuses on enhancing the inference efficiency of large language models (LLMs) through contextual sparsity. While it identifies that contextual sparsity reduces hallucination, it also notes a significant impact on reasoning and deduction performance. To address these drawbacks, the paper introduces SIRIUS, a correction mechanism that effectively improves performance. Specifically, the correction mechanism uses additional parameters (i.e., full model) to check the output of the sparse model based on the input and the same instruction prompt.

**Strengths:**

1. The paper introduces SIRIUS, a correction mechanism that effectively enhances the performance of sparse models, particularly in complex reasoning tasks where traditional sparsity techniques falter.
2. The paper includes many experiments across various datasets and tasks to reveal the problem in the existing contextual sparsity methods and sparse models.

**Weaknesses:**

**Method

1. Based on the results in Tables 1 and 2, this paper claims that “the stronger the model, the larger the quality degradation would be”. However, the studies exclusively utilize relatively smaller models, Llama2-7B and Llama3-8B. Including results from larger LLMs, such as Llama3-70B, would provide a more comprehensive understanding of the impact across different model sizes.

2. As the proposed method uses the probability from the full model to check the correctness of the sparse model, does this require forwarding twice for each token? If so, the cost would be double.

3. It is unclear when to use the correction mechanism. Should it be used after the sparse model has generated all predicted tokens, or immediately after the sparse model predicts each token?

**Experiment

1. In Table 3, if the proposed method uses the full model to correct the output of the sparse model, why is there still a noticeable degradation compared to the full model itself?

2. As the title highlights the efficiency of the proposed method, it would be better to provide some experiments about it like latency (ms) in DejaVu [1].
[1] Deja Vu: Contextual Sparsity for Efficient LLMs at Inference Time. ICML, 2023.

**Minor Issues

There are many typos in this paper, e.g., the title “… Contexual Sparisty …” should be “… Contextual Sparsity …”.

**Questions:**

Please refer to Weaknesses.

**Limitations:**

There are no discussions about the proposed method's limitations and potential societal impact.

---

> ### Author Rebuttal · Authors · 2024-08-07
>
> Thank you so much for your attention and time taken to read the paper and your insightful comments. We are very thankful for you to point out the inadequacy of our method presentation and raising concerns on the diversity of the evaluation. We will try our best to address your concern. If you feel that your concern has been taken into account, please consider raising your score.
>
> 1. Studies mainly use smaller models, which should include Llama-3-70B to verify the claim that "the stronger the model, the larger the quality degradation would be".
>
> In light of your comment, we run the Llama-3-70B-Instruct model on GSM8K, MMLU-FLAN-COT, CoQA, CNN/DailyMail, and TruthfulQA. Due to running Llama-3-70B-Instruct under pipeline parallelism is slow on our hardware (8xA100), MMLU-FLAN-COT is subsampled 10%, and CNN/DailyMail is subsampled 30%.
> |	|GSM-8K-COT|MMLU-FLAN-COT|CoQA|CNN/DailyMail|TruthfulQA|
> |---------|-----------|---------------------------|--------|-------------------|---------------|
> |Llama-3-70B-In|0.9014/0.9022|0.7456|0.6567/0.8069|0.101634/0.020614/0.096413|0.5116/0.4247|
> | + CSparse|0.7407/0.7483|0.7018|0.6497/0.8046|0.101922/0.020854/0.096703|0.4541/0.3807|
> | + FSparse|0.8726/0.8772|0.7193|0.6497/0.8035|0.101505/0.020623/0.096344|0.4835/0.3905|
> |Llama-3-8B-In|0.7612/0.7672|0.6272|0.6153/0.7825|0.101523/0.020481/0.096311|0.4945/0.3647|
> | + CSparse|0.3601/0.3647|0.5307|0.6003/0.7735|0.101681/0.020657/0.096432|0.5067/0.3953|
> | + FSparse|0.6103/0.6202|0.4825|0.5828/0.7577|0.101713/0.020448/0.096516|0.5202/0.3941|
> GSM-8K-COT, “strict match”/”flexible extract”; CoQA, EM/F1; CNN/DailyMail, Rouge-1/2/L, TruthfulQA gen, Rouge-1/2 ACC.
>
> In the table, Llama-3-70B-Instruct still has features that we identified in the paper, sparse excels at prompt understanding (CoQA and CNN/DailyMail). Significant performance degradation occurring at GSM-8K-COT with coarse-grained sparsity. Even though the gap between sparse and dense for 70B is not as big as for smaller models, we don't think it contradicts the claim that the more powerful the model, the more degradation when doing sparse. First, even with Coarse-grained sparsity, the number of parameters is more than 45B, still colossal. Most tasks here are curated in pre-LLM era and are too easy for models now of huge parameters [1]. On the other hand, the claim is presented in the context of comparing Llama-2-7B Chat and Llama-3-8B-Instruct, two smaller but similar models. Llama-2-7B Chat has significantly worse performance in all different tasks. Contextual Sparsity causes much more damage to 8B than 7B. Nevertheless, trying larger parameter models does provide more insights to contextual sparsity characteristics.
>
> [1] Chiang, W. L., Zheng, L., Sheng, Y., Angelopoulos, A. N., Li, T., Li, D., ... & Stoica, I. (2024). Chatbot arena: An open platform for evaluating llms by human preference. arXiv preprint arXiv:2403.04132.
>
> 2. Forwarding twice, cost doubled?
>
> The tokens produced by the sparse model will be fed into the full model for verification, but the latency cost of full model verification will NOT be a lot, compared to the full model generating one new token. The cost in terms of latency won’t be doubled. The full model will verify the tokens generated by the sparse model periodically, usually in a period of 16. During full model verification, all tokens generated by the sparse model that haven’t been verified will be fed in one iteration. And, the full model will generate one more token in this run. Consider the following table of A100 latency of different sequence lengths with bsz=1. Only 1.1ms increase for 64 seqlen. The verification is as light as generating one token, without additional overhead.
> |Input Sequence Length|1|2|4|8|16|32|64|96|
> |-----|-----|-----|-----|-----|-----|-----|-----|-----|
> |Latency (ms)|13.3|13.5|13.6|13.8|14.0|14.9|14.4|17.1|
>
> 3. When to correct?
>
> Correction happens periodically. Specifically, the sparse model would generate PERIOD -1 tokens, and the full model would take in these tokens, verify them based on likelihood thresholds, rollback, and then generate one more token. The cycle goes on. Sirius usually takes PERIOD of 16, sometimes 12.
>
> 4. Degradation after correction?
>
> We can answer by comparing Sirius with Speculative Decoding (SD). SD is lossless. Under the greedy setting, SD acceptance criteria guarantee that the two models' output is the same as the large model output. In comparison, here we effectively loosen up the acceptance criteria. We manually set a threshold on the full model's verification likelihood. i.e., If the full model thinks the token is likely to occur, it accepts. However, the token is not necessarily the one full will greedily decode, leading to divergence in generation and, thus degradation. However, the loosened criteria in turn boost the efficiency of the overall system by a margin to SD's.
>
> 5. Hardware wall clock time speedup?
> Sirius delivers the promised speedup. We present the speedup of Coarse-grained sparsity with Sirius in two settings. First, on-chip, we run Llama-3-8B-Instruct inference. Fine-grained sparsity prior works rely on a custom CUDA kernel, which we don't have access to.
>
> | Setting| GSM-8K-COT | A40 | Speedup | L40 | Speedup | A100 | Speedup |
> |-----|-----|-----|-----|-----|-----|-----|-----|
> | CSparsity | 0.3601 | 20.7ms | 0.66 | 15.6ms | 0.67 | 9.6ms | 0.72 |
> | Sirius (10 kernel size) |0.7309| 24.1ms | 0.78 | 18.2ms | 0.78 | 11.4ms | 0.85 |
> | Sirius (16 kernel size) | 0.7309 | 24.9ms | 0.80 | 18.8ms | 0.81 | 11.8ms | 0.88 |
> | Full | 0.7612 | 30.9ms | | 23.2ms | | 13.3ms | |
>
>
>
> Second, Offloading Llama-3-70B-Instruct. We use a single L40 48GB with a PCIe bus bandwidth to be 25 GB/s.
>
> | | Sparse | Sparse + Sirius | Full |
> |---------------|----------------------|-------------------------------|--------------------|
> | Performance | 0.9014 | 0.8719 | 0.7407 | |
> | Latency (s) | 3.57s | 3.68s | 5.72s | 25.3 GB/s |
> | Ratio to Full | 0.6241 | 0.6434 | | |

---

> ### Author Response · Authors · 2024-08-13
> **Kindly Asking to Reconsider Your Score**
>
> Thank you again for your feedback.
>
> We have added the necessary material to the author's rebuttal to address the points you raised. Given that we only have four reviewers, each reviewer's score significantly impacts the overall assessment of our work. We believe that the current overall assessment does not adequately reflect the contribution of our work. Therefore, we kindly request you reconsider your score. Thank you again for your time and effort in reviewing our paper.

---

> > ### Comment · Reviewer_a8uY · 2024-08-13
> >
> > Thanks for your detailed response. It has addressed some of my concerns. However, I am still concerned that 1) the claim “the stronger the model, the larger the quality degradation would be” since it does not match the experimental results on more SoTA models like Llama-3-70B-In vs. Llama-3-8B-In in the response. 2) The speed-up is slight (e.g., A100: 11.8 ms -> 13.3 ms) compared to the degradation of performance (0.7612 -> 0.7309). Thus, I will keep my score unchanged.

---

> > > ### Author Response · Authors · 2024-08-14
> > > **Response to reviewer**
> > >
> > > We thank the reviewer again for additional feedback and suggestions. We agree that our previous claim “the stronger the model, the larger the quality degradation would be” is not well-stated without enough context and quantifications. Also, the experiments we presented in the first round of rebuttal is not clear.
> > >
> > > We here clarify the claim as: **given the same number of parameters**, the more well-trained (powerful) the model is, the more performance degradation when applying contextual sparsity method would be. In the paper, we present the experiments of Llama-3-8B-Instruct and Llama-2-7B-Chat. Now, we further present the comparison between running Llama-3-70B-Instruct and Llama-2-70B-Chat on GSM8K COT dataset. The experiments are now running and will be soon added. Again, we are deeply grateful for the reviewer’s insightful suggestions on the relationship between performance degradation from contextual sparsity and full model performance.
> > >
> > > On the speedup portion, we would like to point the reader to the speedup ratio between the plain sparse and dense models measured on different devices from the CSparse methods on Llama-3-8B-In (In the table, we show A40, L40, and A100).  Theoretically, the CSparse method only prunes the model by 45% of parameters on Llama-3-8B-In. Therefore, the method will at most achieve 0.65 of the latency as the original dense model, given the LLM inference is bounded by memory. We found that our implementation is very close to optimal for A40 and L40, as the sparse-to-dense ratio on these two devices is close to 0.65 theoretical value, and on top of that, Sirius incurs an additional 10% of the dense model latency as the overhead of correction. **Compared to the sparse model’s original accuracy on GSM-8K-COT 36%, Sirius now corrects it to 73%, which is more than doubled. We argue that Sirius achieves a reasonable efficiency-accuracy tradeoff on A40 and L40.** Moreover, compared to A100, L40 and A40 are much cheaper commodity hardware, which is closer to the users of sparse models that are more resource-limited.
> > >
> > > Admittedly, our current implementation is slightly not as optimal on more high-end GPUs like A100, since the raw sparse-to-dense ratio is close to 0.72 already, which is 0.06 higher than the theoretical value. We found that the slightly suboptimal ratio is caused by extremely high-end GPUs like A100 that are much more demanding on the attention kernels we use. However, building an optimal attention kernel for a relatively short context is beyond the scope of our project. Once the next-generation more optimized attention kernel is rolled out, we will have a similar speedup ratio on A100 as on other slightly slower GPUs. Again, we argue that A100 is much more expensive to rent, making it further away from the target audiences of sparse models.
> > >
> > > Furthermore, we would like to point readers to the accuracy and efficiency tradeoff we have on the Llama-3-70B-In in the offloading setting (part of the weights are loaded on GPU's on-chip memory, the rest is offloaded on CPU RAM), which is one of the only ways a normal practitioner would run inference on the 70B model without high-end GPU clusters. We found that Sirius corrects the CSparse 70B model from 0.76 to 0.87 in accuracy, while only increasing the latency by roughly 2% of the original model latency. (Please note that there is a typo in the offloading 70B table, on the Performance row, the sparse and dense data should be swapped). **Again, we argue that the Sirius method achieves a reasonable tradeoff between Efficiency and Accuracy for the 70B model as well.**
> > >
> > > Thank you again for your additional thoughtful and acute comments. If you think that your concerns have been addressed, please consider adjusting your score.

---

> ### Author Response · Authors · 2024-08-14
> **Follow-up on Previous Response**
>
> We again appreciate the review’s suggestions and inquiries on the efficiency of the claim in our paper. We follow up on our previous message with more experiment results on the comparisons on Llama-2-70B-Chat to provide more empirical studies on the claim. In the table below, we compare two pairs of models on GSM-8K-COT with similar parameters and the degradation after applying the contextual sparse methods (due to time limit, we subsample 20% from the entire dataset for 70B model). Also, we vary the sparsity level to have 50%, 40%, 30% of non-zero values in the weights, where lower than 30% would lead to Llama-3-8B-Instruct to have 0 accuracy).
>
> | Llama-3-70B-Instruct | Accuracy | Degradation | Llama-3-8B-Instruct | Accuracy | Degradation |
> |----------------------|----------|-------------|---------------------|----------|-------------|
> | Full | 0.9205 | | Full | 0.7462 | |
> | Csparse 50% | 0.7652 | 0.1553 | Csparse 50% | 0.3636 | 0.3826 |
> | Csparse 40% | 0.6023 | 0.3182 | Csparse 40% | 0.1856 | 0.5606 |
> | Csparse 30% | 0.3144 | 0.6061 | Csparse 30% | 0.0644 | 0.6818 |
> | Fsparse 50% | 0.8864 | 0.0341 | Fsparse 50% | 0.6477 | 0.0985 |
> | Fsparse 40% | 0.8485 | 0.072 | Fsparse 40% | 0.4053 | 0.3409 |
> | Fsparse 30% | 0.7386 | 0.1819 | Fsparse 30% | 0.0265 | 0.7197 |
> | **Llama-2-70B-Chat** | **Accuracy** | **Degradation** | **Llama-2-7B-Chat** | **Accuracy** | **Degradation** |
> | Full | 0.4508 | | Full | 0.1856 | |
> | Csparse 50% | 0.3939 | 0.0569 | Csparse 50% | 0.1515 | 0.0341 |
> | Csparse 40% | 0.3447 | 0.1061 | Csparse 40% | 0.1098 | 0.0758 |
> | Csparse 30% | 0.2689 | 0.1819 | Csparse 30% | 0.072 | 0.1136 |
> | Fsparse 50% | 0.3864 | 0.0644 | Fsparse 50% | 0.1629 | 0.0227 |
> | Fsparse 40% | 0.3902 | 0.0606 | Fsparse 40% | 0.1364 | 0.0492 |
> | Fsparse 30% | 0.2689 | 0.1819 | Fsparse 30% | 0.1212 | 0.0644 |
>
> From the table, we can clearly see that given similar parameter size, contextual sparsity brings more degradation to Llama-3 family models than to Llama-2 models (Please read the table vertically to compare between Llama-3-70B-Instruct with Llama-2-70B-Chat and Llama-3-8B-Instruct with Llama-2-7B-Chat).
>
>
> Surprisingly, we also notice another interesting phenomenon where the models with larger parameter sizes seem to be more resilient to the contextual sparsity methods. Please read the table horizontally to compare models within the Llama-3 family models and Llama-2 family models. The trend is as expected since the model with a larger parameter size often has more redundancy in parameters.
>
>
> Furthermore, the contextual sparsity’s weakness is in full display in the above table. We can see that for Llama-3-70B-Instruct, even at 50% sparsity, the performance on GSM-8K-COT is comparable to Llama-3-8B-Instruct. Given that the sparse 70B model still has over 40B parameter size, the performance degradation is unacceptable, let alone a sparse 70B model with lower sparsity levels (40% and 30%). Sirius corrects the sparse model to have 87% accuracy and incur negligible overhead during the offloading setting, again showing its effectiveness in helping contextual sparsity methods.
>
>
> Also, as a follow-up on the efficiency-accuracy tradeoff concern, we would emphasize another set of results that was previously overlooked. Please be aware that the accuracy after Sirius correction is dependent on the sparsity level and the sparsity method. Previously, we looked at the coarse-grained sparsity methods where the sparsity pattern is determined for every input prompt and fixed throughout the generation. For the more flexible fine-grained sparsity, where the sparsity pattern changes for different decoded tokens, the sparse method alone achieved 59.68% accuracy on GSM8K-COT, versus the full model’s 75%, Sirius can correct the fine-grained sparse models to 74% accuracy with 0.775 theoretical Average Parameter Used to full model.
>
> Thank you again for your additional thoughtful and acute comments. If you think that your concerns have been addressed, please consider adjusting your score.

---

### Author Rebuttal · Authors · 2024-08-07

We thanked all the reviewers [R1(a8uY), R2(rbBk), R3(ArEw), R4(BB9w)] for their attention and time put into reviewing the paper and also for the thoughtful and supportive comments. We are glad to see that the reviewers find the work interesting [R4(BB9w)] and effective [R1(a8uY)], consider the problem we are solving to be relevant [R2(rbBk)], and think that our proposed technique to be effective [R1(a8uY)]. Also, we are pleased to read that some find our overall presentation to be easy to follow [R2(rbBk)].

On the flip side, we want to make an additional effort to assure the reviewers that we take their suggestions and criticism seriously. Before diving into specific detailed questions, we rewrote the paper submitted considerably based on the precious feedback. The differences can be mainly summarized in the following bullet points. The revised new version of the paper can be accessed with the following link:
https://drive.google.com/file/d/1mvrltX1vd4dlTOKaSyBK1BuXgNvZNYHd/view?usp=sharing.

 - Elaborate on the Difference in Setting between Sirius and Speculative Decoding [R2(rbBk)]
Sirius is a method that seemingly involves two models, a powerful full model and a weaker sparse model. One can easily draw a resemblance with Speculative Decoding, a technique to speed up the large model decoding. The key difference in setting between Sirius and Speculative Decoding is that Speculative Decoding (SD) is LLM-centric, meaning that the key objective is to losslessly speed up the LLM inference. The restriction for whether to accept the weak draft model is high in order to preserve the LLM performance. However, for the users of Contextual Sparse (CS) models, they accept the potential performance degradation involved in the CS models and crave more on the efficiency-performance tradeoff. However, the CS model struggles at difficult tasks that require step-by-step reasoning. Applying Speculative Decoding directly to full and CS models would correct the CS model’s mistake but incur a large price. Sirius is a “small model-centric” technique aiming to still preserve the efficiency of the sparse models (substantially better than SD), but improve sparse model performance in the LLM’s vicinity.

 - Refine the description of the Sirius [R1(a8uY), R2(rbBk), R3(ArEw), R4(BB9w)]
We rewrote the algorithm and the method section in order to present the method in greater detail and address specific concerns about the method relating to the following major questions: When does the switch between full and CS models occur? How is the KV Cache correction implemented? How is the memory of the full with CS models managed, and why we don’t need to load more memory compared to full model inference?

 - Present the Wallclock Time Speedup for Llama-3-8B-Instruct and Llama-3-70B-Instruct Models [R2(rbBk)]
Sirius is about efficiency. We verify the paper’s claim on Average Parameter Used (APU) efficiency with two different high-quality models with different sizes (Llama-3-8B-Instruct and Llama-3-70B-Instruct) on two different settings: on-chip and offloading. Llama-3-8B-Instruct with its CS model can be placed on a single GPU with VRAM greater than 24GB. We implemented a system using Torch Compile and CUDA GRAPH to show that Sirius delivers its promise on three common high-end GPUs, Nvidia A40, L40, and A100. The llama-3-70B-Instruct model has a whopping 140GB of memory requirement in bfloat16, which cannot fit on any single Nvidia high-end graphic card. For normal users with limited resources, they still can run the model with CPU offload. Sirius is also evaluated and delivers the promised speedup.

 - Add Evaluations of Sirius on Diverse Dataset and Finer Ablations on 7 models ranging from 7B to 70B in the Llama family [R3(ArEw)]
We evaluate Sirius in more diverse settings following CS models that identified weaknesses in arithmetic reasoning. Besides GSM8K, we add other arithmetic reasoning datasets AQuA-RAT COT. Also, for commonsense reasoning, we evaluate Sirius' effectiveness on CSQA, StrategyQA, Sports, and Dates. Besides, CS models also struggle at coding, we also evaluate CS models boosted by Sirius on coding datasets HumanEval and MBPP, and show that Sirius can be effective in coding settings.

---

### Author Response · Authors · 2024-08-14
**A Call for AC's Discretion (Thanks to all the reviewer for your attention on Sirius)**

Dear Area Chair,

We are blessed to receive a group of supportive reviewers and an engaging discussion. We believe that we have effectively addressed most of the reviewer's genuine concerns, leading our score to increase from `4345` to `4656`. **However, despite most reviewers looking positively at our work, there is still a lingering concern that our last-minute response doesn’t wait for its feedback.** Thus, we are genuinely looking for AC’s discretion to judge our work’s value. To facilitate AC’s review process on our lengthy discussion, we here provide a one-stop overview and summarize the problem we solve, the method Sirius, the concerns, and how we address them.

**Problem and Sirius**

Contextual Sparsity (CS) refers to the sparsity methods that dynamically change their sparsity pattern during inference based on the context. For LLMs, these methods are especially favorable to static sparsity because CS tends to achieve a higher sparsity level (more efficient) than static methods while maintaining the original full model’s performance. Despite their success on text summarization and conversation tasks, where generation depends on prompts, we show contextual sparsity incurs significant degradation in asks that require LLM’s reasoning and world common sense knowledge.

We present Sirius for the challenge. Since the full weights are natively loaded in GPU memory during CS generation, Sirius efficiently corrects the sparse model by infrequently calling the full model to rewrite the sparse model’s KV Cache, interleaving the full model’s decoded token, and infrequently rolling back unlikely tokens. We show Sirius strong correction with minimal cost through extensive empirical experiments.

**Main Concerns we faced**

*Difference with Speculative Decoding* (`rbBk`)

We appreciate the brilliant question. We show that typical LLMs (Llama-3-8B) tend to achieve a very high acceptance rate (0.89) under the SD setting, but still not enough for the verification period to go past 5 because of the plateauing effect. However, increasing the verification period is the key to amortizing the full model's verification overhead, the longer the period, the more efficient the model is. Sirius can achieve an average advance length of over 13.6 out of a period of 16, compared to SD's less than 9, bringing the average parameter used per token to be lower than 0.76 compared to 0.86 minimum by SD.

*Practical Speedup from Sirius* (`a8uY` and `rbBk`)

To address the concern that Sirius lacks practical speedup in the paper. We further implement the system for two mainstream models (Llama-3-8B and Llama-3-70B) under two different settings (on-chip and offloading). We show that for 8B model under the on-chip setting, we correct the performance from 36% to 73% for GSM8K-COT with a latency ratio from 0.65 to 0.78. For 70B under offloading, we correct from 76% to 87% with a latency ratio from 0.62 to 0.64. Both experiments show that Sirius maintains a strong tradeoff between performance and accuracy.

*Miscommunication on Sirius method* (`a8uY`,`ArEw`,`BB9w`)

We carefully answer the reviewer's confusion and take in their suggestions in our further refined version of the paper. Reviewers raising the concern reflect that their concerns are addressed during further exchange.

*Lack of empirical evaluations* (`ArEw`)

We further evaluate Sirius across 6 different models on 7 different tasks ranging from arithmetic reasoning, commonsense reasoning, and coding. The experiments have shown that Sirius is effective in general settings. The review expressed the concern is effectively "alleviated".

*Given the similar parameter count, the more well-trained the model is, the more degradation would be* (`a8uY`)

The original claim isn't clearly defined, which is corrected: given the similar parameter count, the more well-trained (powerful) the model is, the more degradation contextual sparsity would bring to the sparse model. To support the claim, we present experiments comparing Llama-2-7B to Llama-3-8B and Llama-2-70B to Llama-3-70B. We show clearly that given the similar parameter size, Llama-3 models that generally score higher than Llama-2, lose accuracy much more from CS under different sparsity levels. We believe that the experiment concretely validates the claim. Also, the observation makes the challenge more urgent when models now become more well-trained given the same parameter size. We deeply appreciate this concern raised.

**However, since the results are presented at the last minute, we worry that the reviewer doesn't have enough time to go through and provide feedback.**

---

We wish that the overview we presented is concise for reviewers and AC. Also, we hope that the overlooked and important problem our work revealed will be attended to by the efficiency community, and Sirius will be the pioneering step in bridging the gap between contextually sparse models and the full models.

Sincerely,

Paper20976 Authors

---

### Decision · Program_Chairs · 2024-09-25

**Decision:**

Accept (poster)

**Comment:**

This paper investigates the contextual sparsity, by first studying the success and failure patterns of applying contextual sparsity method to modern LLMs. The study indicates that the contextual sparsity method often fails in reasoning and deduction tasks like GSM8K. The authors then propose a novel approach, SIRIUS, to improve the contextual sparsity according to the observation. The approach employed speculative decoding like technique to self-correct the decoding process. Experiments show the proposed change leads to a significant improvement in the GSM8K task while maintain the performance on other tasks.

The authors are very engaged in the rebuttal process, although there are still some concerns like the improvement is relative light compared to the performance drop, most reviewers are positive about the paper. And the study of it is indeed motivated and the results supports most of the claims.